# Identification of an orally active small-molecule PTHR1 agonist for the treatment of hypoparathyroidism

Tatsuya Tamura[1], Hiroshi Noda[1], Eri Joyashiki[1], Maiko Hoshino[1], Tomoyuki Watanabe[1], Masahiko Kinosaki[1], Yoshikazu Nishimura[1], Tohru Esaki[1], Kotaro Ogawa[1], Taiji Miyake[1], Shinichi Arai[1], Masaru Shimizu[1], Hidetomo Kitamura[1], Haruhiko Sato[1] & Yoshiki Kawabe[1]

Parathyroid hormone (PTH) is essential for calcium homeostasis and its action is mediated by the PTH type 1 receptor (PTHR1), a class B G-protein-coupled receptor. Hypoparathyroidism and osteoporosis can be treated with PTH injections; however, no orally effective PTH analogue is available. Here we show that PCO371 is a novel, orally active small molecule that acts as a full agonist of PTHR1. PCO371 does not affect the PTH type 2 receptor (PTHR2), and analysis using PTHR1–PTHR2 chimeric receptors indicated that Proline 415 of PTHR1 is critical for PCO371-mediated PTHR1 activation. Oral administration of PCO371 to osteopenic rats provokes a significant increase in bone turnover with limited increase in bone mass. In hypocalcemic rats, PCO371 restores serum calcium levels without increasing urinary calcium, and with stronger and longer-lasting effects than PTH injections. These results strongly suggest that PCO371 can provide a new treatment option for PTH-related disorders, including hypoparathyroidism.

---

[1] Research Division, Chugai Pharmaceutical Co., Ltd., 1-135, Komakado, Gotemba, Shizuoka 412-8513, Japan. Correspondence and requests for materials should be addressed to T.T. (email: tamuratty@chugai-pharm.co.jp).

Parathyroid hormone (PTH) is an 84 amino acid peptide that is secreted from the parathyroid glands in response to low blood calcium (Ca) levels. PTH is the principal regulator of blood Ca concentration, acting primarily on bones and kidneys[1,2]. Biological actions of PTH are mediated via stimulation of the PTH type 1 receptor (PTHR1), a member of the class B family of G-protein-coupled receptors (GPCRs). PTHR1 couples strongly to the adenylate cyclase-cyclic AMP (cAMP)-protein kinase A signalling pathway, and generally less robustly to the phospholipase C–protein kinase C–intracellular $Ca^{2+}$ signalling pathway[3]. Activation of PTHR1 can also promote recruitment of β-arrestins, leading to internalization of PTHR1 as well as to stimulation of intracellular signalling pathways[4,5]. Although PTH injections are clinically available, orally available agonists of PTHR1 would offer several advantages over existing therapies. Whereas small molecules that activate class A GPCRs are used in clinical practice[6,7], a small-molecule agonist for class B GPCRs, which include PTHR1, is not yet available.

Lack of functional PTH results in hypoparathyroidism, which is a rare disease characterized by hypocalcemia. The clinical manifestations of hypoparathyroidism include muscle cramping and potentially life-threatening complications, such as seizure or laryngeal spasm[8,9]. Conventional therapy for hypoparathyroidism involves pharmacological doses of oral Ca and active vitamin D. However, although this treatment regimen can increase intestinal absorption of Ca, it cannot restore renal Ca reabsorption, and can thereby lead to hypercalciuria. Chronic hypercalciuria can cause irreversible renal damage and eventual renal failure[8,10,11]. To avoid such risk, studies have focused on PTH replacement therapy using the full-length, native PTH molecule called human PTH(1–84) (hPTH(1–84))[12,13] or a synthetic fragment called human PTH(1–34) (hPTH(1–34))[8,14,15]. PTH replacement therapy has been demonstrated to ameliorate the blood, renal, skeletal and neuropsychological features of hypoparathyroidism to a greater extent than conventional treatment. hPTH(1–84) and hPTH(1–34) can elevate serum Ca without raising urinary Ca in patients with hypoparathyroidism; however, due to the short half-lives of these compounds, frequent injections or continuous pump delivery is necessary to maintain serum Ca levels at a steady level[12,13,15]. The peptidic nature (or poor oral bioavailability) of PTH(1–34) has limited its use in the treatment of osteoporosis, since daily subcutaneous injection of hPTH(1–34) is also used clinically for treating severe osteoporosis[1,16]. Given the chronic nature of hypoparathyroidism and osteoporosis, orally active PTH mimetics are desirable and will benefit patients in their long-term treatment. However, no orally active compounds for PTHR1 have been identified so far[7,17,18].

In this study, we identified a nonpeptidyl small-molecule PTHR1 agonist, PCO371, through a cell-based functional screening assay. The compound activates human PTHR1 (hPTHR1) but not the human PTH type 2 receptor (hPTHR2). Analysis using hPTHR1–hPTHR2 chimeric receptors demonstrated that Proline 415 in transmembrane segment 6 (TM6) of hPTHR1 is a key residue for the receptor selectivity and activation. Orally administered PCO371 exhibits PTH-like biological activity in osteopenic ovariectomized (OVX) rats and in hypocalcemic thyroparathyroidectomized (TPTX) rats. In the former model, intravenously administered PCO371 increases bone mineral density (BMD) and bone strength as effectively as PTH; however, when administered orally, only a limited increase in BMD and bone strength is observed. In the latter model, 4-week oral administration shows ability to restore the normal serum Ca level without increasing urinary Ca. To our knowledge, PCO371 is the first example of a small-molecule PTHR1 agonist with good oral bioavailability and efficacy in animal models.

The safety and pharmacokinetics of PCO371 are currently being evaluated in humans.

## Results

***In vitro* characterization of PCO371**. PCO371 (Fig. 1a) was identified through a cell-based functional screening assay with transfected cells expressing hPTHR1. In COS-7 cells expressing hPTHR1, PCO371 stimulated cAMP production in a dose-dependent manner ($EC_{50} = 2.4 \, \mu mol \, l^{-1}$, Fig. 1b). Similar results were obtained in COS-7 cells expressing rat PTHR1 (Supplementary Fig. 1a). PCO371 also enhanced the phospholipase C activity ($EC_{50} = 17 \, \mu mol \, l^{-1}$) as did hPTH(1–34) (Fig. 1c). PCO371 stimulated cAMP production in COS-7 cells transfected with hPTHR1-delNT, which lacks the N-terminal extracellular domain of hPTHR1 (ref. 19), with an $EC_{50}$ value ($EC_{50} = 2.5 \, \mu mol \, l^{-1}$) similar to that of cells expressing full-length hPTHR1. On the other hand, the ability of hPTH(1–34) to induce cAMP production was significantly lower in cells transfected with hPTHR1-delNT than in those with the full-length hPTHR1 (Fig. 1b,d). We therefore examined whether PCO371 binds to the transmembrane domain of PTHR1 using cell membranes prepared from COS-7 cells expressing hPTHR1 and using $^{125}I$-[Aib$^{1,3}$,Nle$^{8}$, Gln$^{10}$,Har$^{11}$,Ala$^{12}$,Trp$^{14}$,Tyr$^{15}$]–PTH(1–15) [$^{125}I$-[Aib$^{1,3}$,M]–PTH(1–15)], which interacts mainly with the receptor's transmembrane domain, as a radiolabelled tracer[20]. PCO371 at a concentration of $100 \, \mu mol \, l^{-1}$ inhibited the binding of $^{125}I$-[Aib$^{1,3}$,M]–PTH(1–15) almost completely (Fig. 1e), suggesting that PCO371 exerts its agonist activity by interacting with PTHR1's transmembrane domain.

To assess the selectivity of PCO371 for PTHR1, we first examined the reactivity of PCO371 with PTHR2, which also belongs to the class B GPCRs and is most abundantly expressed in the central nervous system. hPTHR2 shares ∼50% amino acid sequence identity with hPTHR1, and ∼82% amino acid sequence identity with rat PTHR2 (ref. 21). PCO371 did not stimulate cAMP production in transfected COS-7 cells expressing human or rat PTHR2, while tuberoinfundibular peptide of 39 residues (TIP39), the cognate ligand for PTHR2, showed an increase in cAMP production (Fig. 1f and Supplementary Fig. 1b). The specificity of PCO371 to hPTHR1 was further examined in functional GPCR cell-based assays to determine agonistic and antagonistic activities against 12 class B GPCRs. PCO371 did not exhibit activity at 1 or $10 \, \mu mol \, l^{-1}$, whereas the cognate ligands and relevant reference compounds showed significant activity in these assays (Supplementary Table 1).

We also examined whether PCO371 induces cAMP production via endogenously expressed PTHR1 by using UMR-106 cells (a rat osteosarcoma cell line) that express endogenous rat PTHR1. Although the potency was much weaker than that of hPTH(1–34), PCO371 induced cAMP production as did hPTH(1–34), suggesting that PCO371 acts as a full agonist of PTHR1 (Fig. 2a).

PTHR1 has the capacity to form two distinct high-affinity conformational states: one is G-protein-uncoupled conformation ($R^0$), and the other is G-protein-coupled conformation ($RG$)[22]. Whereas a PTH-related peptide binds weakly to $R_0$ and exhibits only a transient response[22–24], PTH-related peptide analogues, such as M-PTH(1-28), M-PTH(1–34) and LA-PTH, bind with greater affinity to $R^0$ and produce prolonged calcemic responses *in vivo*. The peptides that bind $R^0$ with high affinity are able to produce cAMP signalling responses in PTHR1-expressing cells for a certain amount of time after initially binding to PTHR1. A previous study also suggested that there is a good correlation between $R^0$ binding affinity and the duration of the cAMP response induced by a given PTH ligand after initial binding to

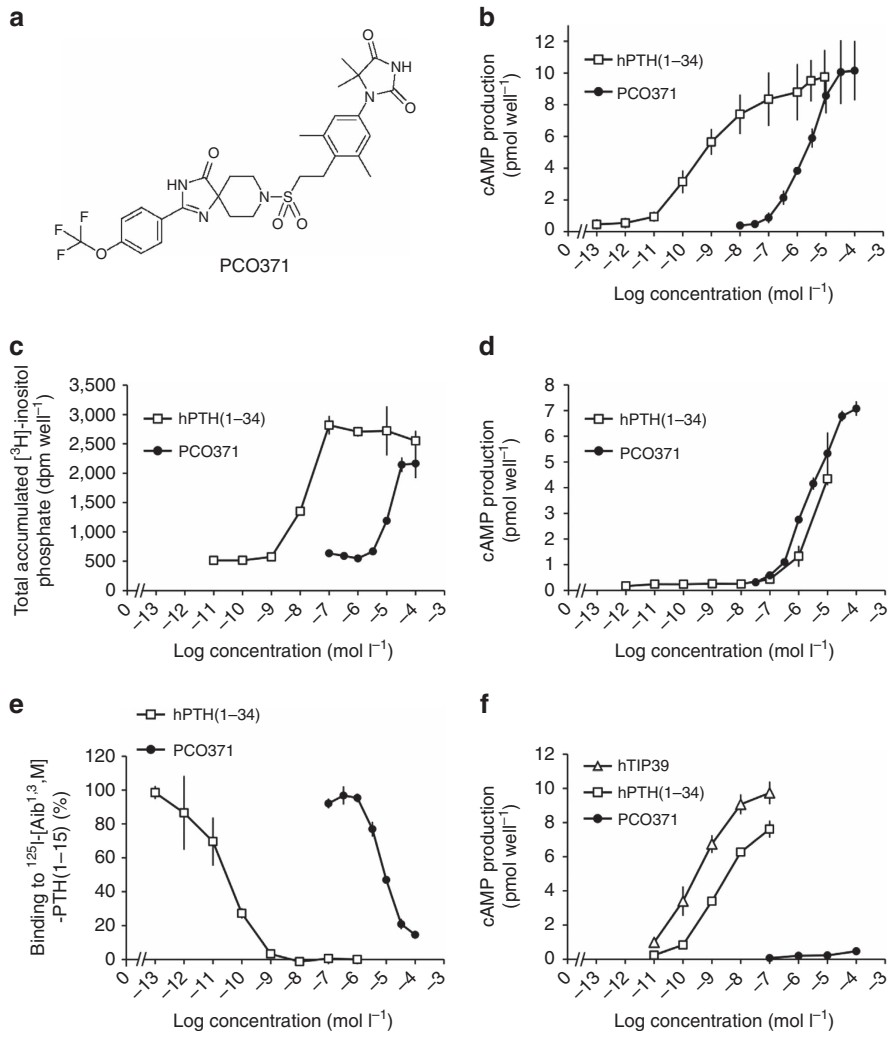

**Figure 1 | Effects of PCO371 on COS-7 cells transfected with hPTHR1 or hPTHR2.** (**a**) Chemical structure of PCO371: 1-{3,5-dimethyl-4-[2-({4-oxo-2-[4-(trifluoromethoxy)phenyl]-1,3,8-triazaspiro[4.5]dec-1-en-8-yl} sulfonyl)ethyl]phenyl}-5,5-dimethylimidazolidine-2,4-dione. (**b**) cAMP production by hPTH(1–34) and PCO371 in COS-7 cells expressing hPTHR1. (**c**) phospholipase C (PLC) activity (as indicated by accumulation of total [³H]-inositol phosphate production) induced by hPTH(1–34) and PCO371 in COS-7 cells expressing hPTHR1. No response to either agonist was detected in COS-7 cells transfected with vector alone. (**d**) cAMP production by hPTH(1–34) and PCO371 in COS-7 cells expressing N-terminal-deleted hPTHR1 (hPTHR1-delNT). (**e**) Competition with ¹²⁵I-[Aib¹,³,M]-PTH(1–15) in COS-7 cells expressing hPTHR1. (**f**) cAMP production by hPTH(1–34), PCO371 and hTIP39 in COS-7 cells expressing hPTHR2, the cognate receptor for hTIP39. Data are represented as the mean ± s.d. of three experiments in duplicate (n = 6 for **b**) and one experiment in triplicate (n = 3 for **c**–**f**).

PTHR1 (ref. 23). We therefore examined the sustained cAMP signalling responses by PCO371 or LA-PTH in comparison with hPTH(1–34) in a cAMP washout assay in UMR-106 cells. The duration of cAMP-signalling response induced by PCO371 was much shorter than that of hPTH(1–34), whereas LA-PTH, a long-acting PTH analogue, showed more prolonged cAMP signalling than hPTH(1–34) (Fig. 2b).

In UMR-106 cells, PCO371 also dose-dependently increased expression of bone-related messenger RNAs (mRNAs) for Fos (encoded by *Fos*), osteocalcin (encoded by *Bglap*) and receptor activator of nuclear factor-κB ligand (RANKL, encoded by *Tnfsf11*), and dose-dependently decreased expression of mRNAs for osteoprotegerin (encoded by *Tnfrsf11b*) and sclerostin (encoded by *Sost*), as did hPTH(1–34) (Fig. 2c–g). In these assays, 1 µmol l⁻¹ of PCO371 was able to cause changes in mRNA expressions in UMR cells even when PCO371 did not induce cAMP production in the cells (Fig. 2a). This anomaly may be due to the difference in incubation time (6 h for gene expresssions versus 20 min for cAMP production) or due to a level of cAMP elevation that despite being undetectable has the ability to transmit signals in cells²⁵⁻²⁷.

We next examined whether PCO371 induces PTH-like bone-resorbing activity in organ cultures with fetal rat long bones. PCO371 at 1 µmol l⁻¹ or more had the ability to stimulate ⁴⁵Ca release from prelabeled bones as did hPTH(1–34) (Fig. 3).

**Mechanism of selectivity for PTHR1 over PTHR2.** To investigate the mechanisms underlying the selectivity of PCO371 for hPTHR1 over hPTHR2, we performed cAMP production assays using transfected COS-7 cells expressing hPTHR1 mutants in which amino acid residue(s) were replaced by the corresponding hPTHR2 amino acid residue(s) (Fig. 4a,b). hPTH(1–34) increased cAMP in cells expressing hPTHR1–hPTHR2 chimeric receptors (C1, C2 or C3), whereas PCO371 did not do so in cells expressing C1 or C2 (Fig. 4c). Since the common replaced segment in C1 and C2 mutants is TM regions 6 and 7, the critical role of these

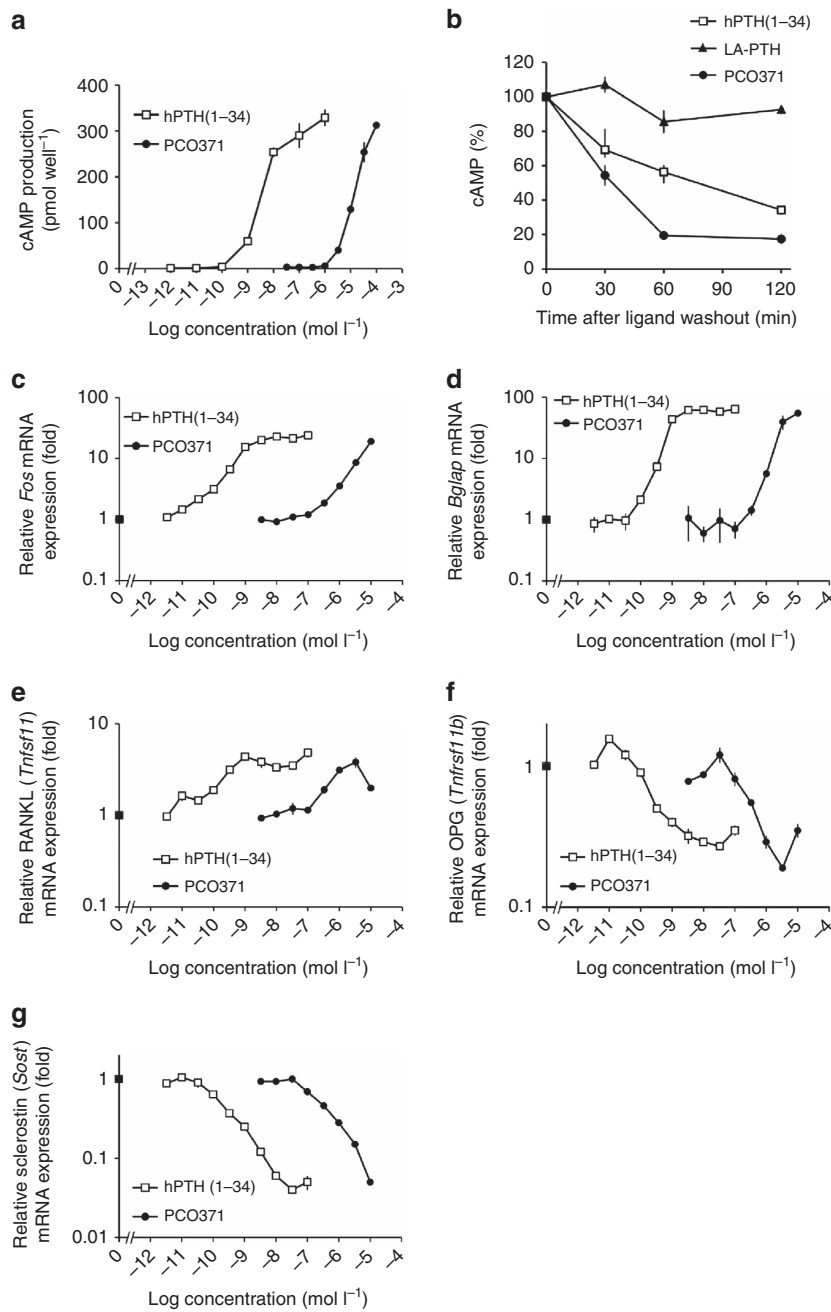

**Figure 2 | Effects of PCO371 on PTHR1 signalling in UMR-106 cells.** (**a**) cAMP production by hPTH(1–34) and PCO371 in UMR-106 cells, which natively express rat PTHR1. (**b**) Duration of cAMP-signalling responses induced by PCO371, hPTH(1–34) and LA-PTH in UMR-106 cells. The time courses of cAMP responses produced by PCO371 (0.1 mmol l$^{-1}$), hPTH(1–34) (1 μmol l$^{-1}$) or LA-PTH (0.1 μmol l$^{-1}$) in UMR-106 cells were examined by cAMP washout assay. The cAMP response is expressed as a percentage of the maximal cAMP produced in the cells treated with each ligand (determined by incubating cells concomitantly with the ligand for 10 min, and further incubating with IBMX for 5 min without a washing-out phase). The range of maximum cAMP values was PCO371: 17.4 + 0.2 pmol per well, hPTH(1–34): 21.3 + 0.9, LA-PTH: 42.2 + 0.9, and basal cAMP: 1.8 + 0.2. (**c–g**) Effects of PCO371 and hPTH(1–34) on the expression of mRNAs for c-fos (*Fos*) (**c**), osteocalcin (*Bglap*) (**d**), RANKL (*Tnfsf11*) (**e**), osteoprotegerin (*Tnfrsf11b*) (**f**) and sclerostin (*Sost*) (**g**). mRNA levels are shown as fold-change over control. Gene expression was analysed at 1 h (**c**) or 6 h (**d–g**) after treatment of UMR-106 cells with hPTH(1–34) or PCO371. All genes were normalized to 18 S ribosomal RNA. Data are represented as the mean ± s.d. of one experiment (*n* = 3 for **a–g**).

regions is indicated. Furthermore, because cells expressing C3 responded well to PCO371, we examined the effect of replacing the amino acid residues in the TM6 and TM7 intracellular region with those of hPTHR2 (mutant receptors M24, M25 and M26; Fig. 4b). A lack of cAMP increase by PCO371 was shown only in mutant receptor M25, which contains three mutated amino acids (Methionine 414, Proline 415 and Lysine 416; Fig. 4c). In subsequent experiments with single amino acid replacement, Methionine 414 to Valine (M414V) and Leucine 416 to Valine (L416V) had only modest effects on cAMP response, whereas Proline 415 to Leucine (P415L) resulted in a dramatic reduction of cAMP production by PCO371 (Fig. 4c,d). These results indicate that Proline 415 is essential for the selective activation of PTHR1 by PCO371.

**Pharmacokinetics profile in normal rats**. Pharmacokinetics profiles of PCO371 and hPTH(1–34) after single administration to normal rats are summarized in Table 1. Exposures following oral dosing of PCO371 dose-dependently increased within the dose range tested. The maximum plasma concentrations ($C_{max}$) of PCO371 were attained at 1–1.5 h, and the terminal half-lives ($T_{1/2}$) were 1.5–1.7 h. Oral bioavailability of PCO371 was 34% at a dose of $2 \, mg \, kg^{-1}$. After a single subcutaneous injection of hPTH(1–34), the concentration in serum was elevated within 15 min in a dose-dependent manner, and the $T_{1/2}$ was 15 min.

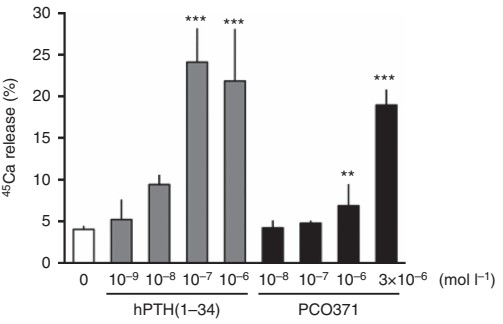

**Figure 3 | Effects of PCO371 on $^{45}$Ca release from pre-labelled fetal rat long bones.** Control medium contained only vehicle for PCO371 (0.1% DMSO) or hPTH(1–34) ($10 \, \mu mol \, l^{-1}$ acetic acid). Data are represented as the mean + s.d. of one experiment ($n = 4$). Williams's test was used to compare PCO371- or hPTH(1–34)-treated groups with vehicle-treated group; $^{**}P < 0.01$, $^{***}P < 0.001$.

***In vivo* experiments in OVX rats**. To evaluate the effect of PCO371 on bone metabolism, we examined its effects on BMD, bone strength and serum biochemistry in mature OVX rats. Treatment began 3 months after ovariectomy when the BMD of lumbar vertebrae in OVX rats was significantly reduced compared with sham-operated rats. The anabolic and catabolic actions of PTH depend on the dosage regimen, and intermittent, not continuous, administration of PTH increases bone mass[28]. We therefore examined the effect of PCO371 on OVX rats after oral or intravenous administration, which have different pharmacokinetics profiles (Supplementary Fig. 2 and Supplementary Table 2). Twelve-week treatments of once-daily intravenous PCO371 ($10 \, mg \, kg^{-1}$) or subcutaneous hPTH(1–34) injections caused significant increase in BMD and bone strength in the lumbar vertebrae of OVX rats compared with OVX rats dosed with vehicle (Fig. 5a,c). Whereas daily oral PCO371 ($30 \, mg \, kg^{-1}$) and intravenous PCO371 ($3 \, mg \, kg^{-1}$) did not affect BMD or bone strength in the lumbar spine (Fig. 5a,c), daily oral PCO371 ($30 \, mg \, kg^{-1}$) partially increased BMD in the proximal femur (Fig. 5b), but not in total femur (Supplementary Fig. 3). Both oral and intravenous ($10 \, mg \, kg^{-1}$) PCO371 induced a significant increase in serum osteocalcin and urinary collagen type 1 cross-linked C-telopeptide (CTX)/creatinine (Fig. 5d,e), which suggests accelerated bone turnover, and this was supported histologically by an increase in bone formation (Fig. 5f and Supplementary Table 3) and bone resorption parameters in the lumbar vertebrae (Supplementary Table 3). Serum Ca was within the normal range in all rats treated with PCO371 or hPTH(1–34) (Fig. 5g).

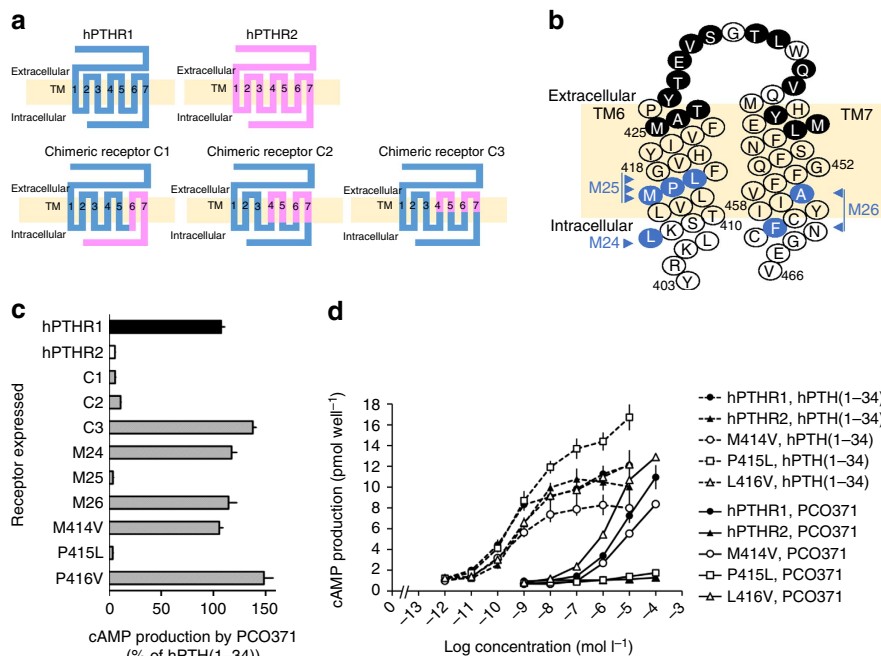

**Figure 4 | Mutation studies using chimeric receptors of hPTHR1 and hPTHR2.** (**a**) Schematic structures of hPTHR1, hPTHR2 and chimeric receptors in which blue indicates amino acids comprising hPTHR1 and pink indicates amino acids comprising hPTHR2. C1 is a chimeric hPTHR1 in which the C-terminal region (from 398 to 593) is replaced by the corresponding region of hPTHR2 (from 353 to 550). C2 is a chimeric hPTHR1 in which two regions (from 325 to 390 and from 407 to 461) are replaced by the corresponding regions of hPTHR2 (from 280 to 345 and from 362 to 415). Similarly, in C3 two regions (from 334 to 378 and from 425 to 446) are replaced by the corresponding regions of hPTHR2 (from 289 to 333 and from 380 to 400). (**b**) Snake diagram of TM regions 6 and 7 of hPTHR1 based on the topological arrangement of class B GPCRs (ref. 4). The amino acid residues of hPTHR1 that were replaced with the corresponding residues of hPTHR2 in C3 (filled black circles) and in mutant receptors M24 (L407), M25 (M414, P415 and L416), M26 (A456 and F461), M414V, P415L and L416V (filled blue circles) are indicated. (**c**) cAMP production stimulated by PCO371 ($0.1 \, mmol \, l^{-1}$) in COS-7 cells expressing hPTHRs or mutant hPTHR1s, indicated as a percentage of cAMP production stimulated by $0.1 \, \mu mol \, l^{-1}$ hPTH(1–34). Data are represented as the mean + s.d. of one experiment ($n = 3$) (**d**) Stimulation of cAMP production by hPTH(1–34) and PCO371 in cells expressing hPTHR1, hPTHR2 and mutant hPTHR1s. Data are represented as the mean ± s.d. of one experiment ($n = 3$).

**Table 1 | Pharmacokinetics parameters of hPTH(1–34) and PCO371 in normal rats.**

| | Route | Dose | $T_{1/2}$ (h) | $T_{max}$ (h) | $C_{max}$ (ng ml$^{-1}$) | AUC$_{inf}$ (ng h ml$^{-1}$) | BA % |
|---|---|---|---|---|---|---|---|
| hPTH(1–34) (nmol kg$^{-1}$) | s.c. | 3 | 0.3 ± 0.1 | 0.1 ± 0.1 | 2.42 ± 0.53 | 0.99 ± 0.17 | 22 |
| PCO371 (mg kg$^{-1}$) | p.o. | 2 | 1.5 ± 0.1 | 1.5 ± 0.9 | 92.5 ± 37.1 | 374 ± 187 | 34 |
| | | 6 | 1.4 ± 0.2 | 1.0 ± 0.0 | 727 ± 103 | 2,360 ± 280 | NE |
| | | 18 | 1.7 ± 0.3 | 1.3 ± 0.6 | 4,560 ± 710 | 18,600 ± 1,900 | NE |

NE, not examined; BA, bioavailability.
Rats were treated with a single subcutaneous injection of hPTH(1–34), or a single oral administration of PCO371. Blood samples were collected at 2, 5, 10, 15, 30 and 45 min, and 1 and 2 h [for hPTH(1–34)] or at 15 and 30 min, and 1, 2, 4, 6, and 24 h (for PCO371), and pharmacokinetics parameters were determined. Data are represented as the mean ± s.d. of one experiment (n = 3 for PCO371, n = 4 for hPTH(1–34)).

**In vivo experiments in TPTX rats**. We next examined the effect of PCO371 on serum Ca and inorganic phosphate (P$_i$) in TPTX rats. After a single oral administration, PCO371 dose-dependently increased serum Ca and decreased P$_i$, with greater efficacy and longer-lasting effects than those of hPTH(1–84) (Fig. 6a,b) or hPTH(1–34) (Supplementary Fig. 4a,b). Pharmacokinetics profiles of PCO371 and hPTH(1–84) were determined in TPTX rats. PCO371 had longer serum $T_{1/2}$ and $T_{max}$ than hPTH(1–84) (Supplementary Table 4). We then examined the efficacy of PCO371 in 4-week multiple oral dosing studies in TPTX rats. In this model, subcutaneous treatment with hPTH(1–84) or hPTH(1–34) transiently increased serum Ca to within the target therapeutic range (7.6–11.2 mg dl$^{-1}$)[10,11], but the serum Ca level dropped quickly to the basal level following each injection (Supplementary Fig. 4c,d). No increase in urinary Ca excretion was observed after 4-week treatment of hPTH(1–84) (Supplementary Fig. 4e). Twice-daily oral PCO371 at 3 mg kg$^{-1}$ or more dose-dependently increased serum Ca to within the target therapeutic range, and maintained the Ca level within this range for at least 6 h after each administration from the first administration until the end of the study (Fig. 6c). In this experiment, no hypercalcemia ( > 11.2 mg dl$^{-1}$) was observed at 0, 6 and 10 h after administrations of PCO371 on days 1, 7, 14, 22 and 29. Urinary Ca excretion after 4-week treatment was not increased by treatment with PCO371 up to 9 mg kg$^{-1}$, even when the serum Ca had reached the target therapeutic range (Fig. 6d), possibly because renal Ca reabsorption was stimulated. Serum 1α,25-dihydroxyvitamin D$_3$ [1,25(OH)$_2$D$_3$] levels increased significantly in rats treated with 9 mg kg$^{-1}$ PCO371 (Fig. 6e), similarly to animals treated with hPTH(1–34) (Supplementary Fig. 4f). There was no significant difference in body weight or general condition between vehicle-treated rats and PCO371-treated rats (Supplementary Fig. 4g).

An important benefit of PTH replacement therapy when treating hypoparathyroidism is that it can reduce the doses of Ca and vitamin D$_3$ while maintaining serum Ca levels, and this has been clinically demonstrated by combining hPTH(1–84) with conventional vitamin D and Ca therapy[12]. We therefore examined the effect of PCO371 in combination with alfacalcidol (1α-hydroxycholecalciferol) in TPTX rats. Alfacalcidol alone at 75 ng kg$^{-1}$ normalized the serum Ca to within or above the target therapeutic range, but was accompanied by considerable urinary Ca excretion (Fig. 6f,g). Once-daily oral add-on treatment with 6 mg kg$^{-1}$ of PCO371 plus 38 ng kg$^{-1}$ of alfacalcidol increased serum Ca to levels comparable with that by 75 ng kg$^{-1}$ of alfacalcidol alone (Fig. 6f). The combination of PCO371 plus alfacalcidol did not elevate urinary Ca excretion, indicating that PCO371 can enable the dose of alfacalcidol to be lowered, thereby reducing urinary Ca excretion (Fig. 6g). These results indicate that PCO371 can be useful for the treatment of hypoparathyroidism both in monotherapy and as add-on therapy to conventional therapy.

## Discussion

As a principal regulator of Ca homeostasis, PTH increases tubular reabsorption of Ca in the kidney and promotes renal excretion of phosphate. In blood, PTH stimulates activity of 25-hydroxyvitamin D$_3$-1α-hydroxylase, which increases production of 1,25(OH)$_2$D$_3$, which then stimulates intestinal absorption of Ca; whereas in bone, PTH stimulates bone turnover and mobilizes Ca into the circulation[1,2]. Accordingly, insufficient levels of PTH can cause hypocalcemia. Patients with hypoparathyroidism require very high doses of vitamin D and Ca supplements to normalize serum Ca; however, this treatment risks hypercalciuria and may lead to nephrocalcinosis and subsequent loss of renal function. To reduce (or prevent) hypercalciuria, thiazide diuretics can be used to lower urinary Ca (ref. 14). Recently, PTH replacement therapy with recombinant hPTH(1–84) has been approved to treat hypoparathyroidism; however, as a peptide it requires daily subcutaneous injections[12,29]. Once-daily dosing of hPTH(1–84) can increase Ca to peak levels at around 7 h after administration, and the effects are observed for up to 24 h. However, hypercalcemia was observed at peak times in 71% of hPTH(1–84)-treated patients, and this suggests that a lower dose or more frequent daily dosing regimens may be more beneficial for some patients[30]. Studies with hPTH(1–34) have also demonstrated that pump delivery or twice-daily delivery of the peptide provides higher physiological Ca levels than once-daily injections[15].

The present study demonstrates that PCO371 exhibits PTH-like activity with regard to Ca homeostasis, such as renal Ca reabsorption, 1,25(OH)$_2$D$_3$ production, and Ca release from the bone (Figs 3 and 6a,c–e). PCO371 was also able to mimic the effects of PTH on bone metabolism (Fig. 5a–f and Supplementary Table 3) and phosphate homeostasis (Fig. 6b) in vivo, which demonstrates the compound's potential as a PTH mimetic. Although the in vitro activity of PCO371 was 1,000- to 10,000-fold less potent than that of hPTH(1–34), oral PCO371 exhibited more robust and longer-lasting calcemic effects than did subcutaneous hPTH(1–34) or hPTH(1–84) in vivo. The duration of cAMP-signalling response induced by PCO371 in UMR-106 cells was not prolonged compared with PTH(1–34) or LA-PTH (Fig. 2b). These results suggest that the superior calcemic activity in response to PCO371 in vivo presumably stems from its good bioavailability and longer half-life than that of the hPTHs. Differences between in vitro activity and in vivo response, which are mainly due to the difference in pharmacokinetics, were found in earlier studies with PTH peptide analogues[18].

In the present study, we used TPTX rats as an animal model for hypoparathyroidism, and demonstrated that complementing alfacalcidol therapy with PCO371 increased serum Ca to a level comparable with that produced by twice the dose of alfacalcidol without elevating urinary Ca excretion. Our results suggest that replacing conventional therapy for hypoparathyroidism with

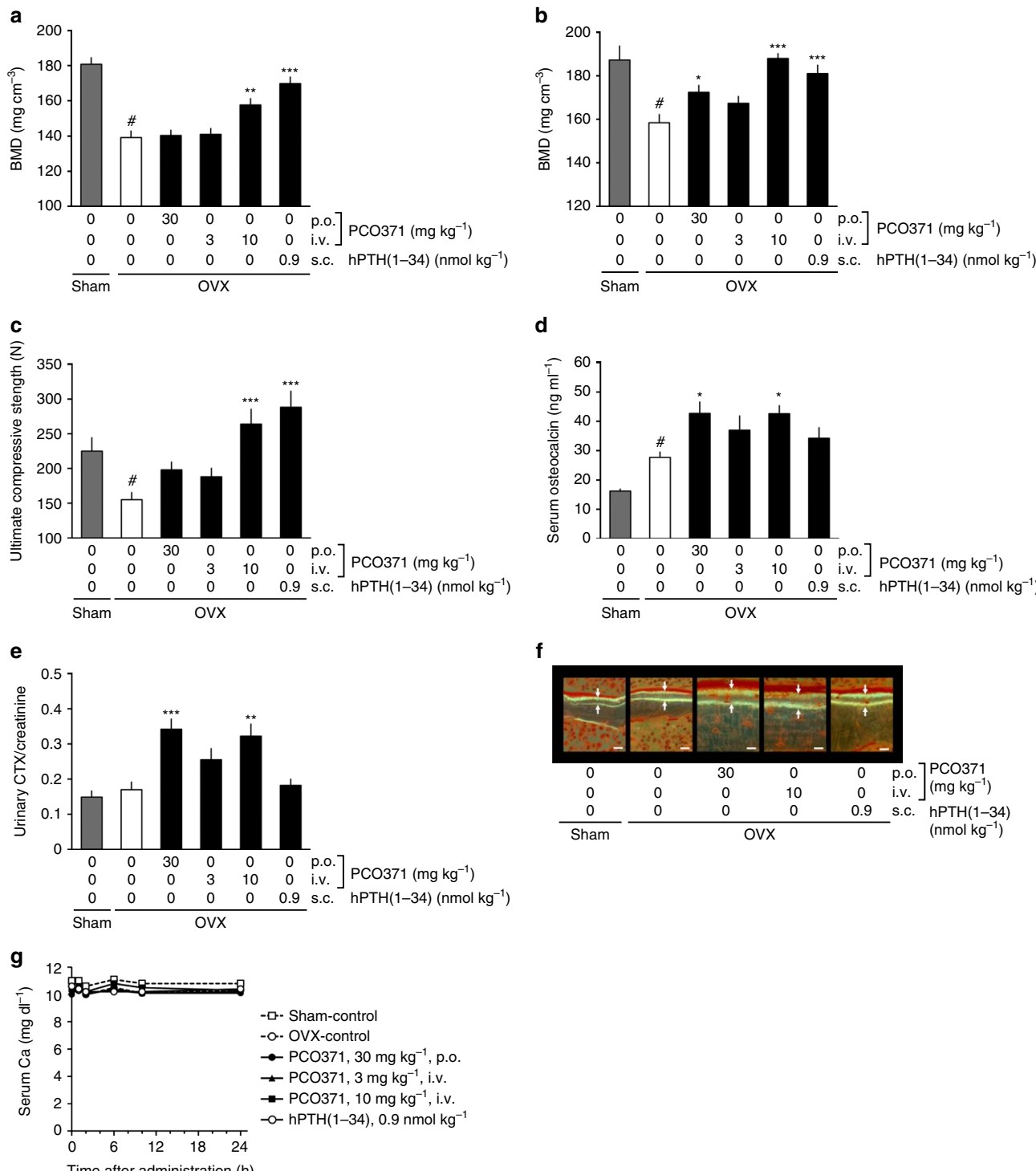

**Figure 5 | Effects of PCO371 on bone parameters in OVX rats.** Rats were treated once-daily with PCO371 (p.o.) or hPTH(1–34) (s.c.) for 12 weeks commencing at 12 weeks after OVX surgery. (**a,b**) BMD of the lumbar spine (L3–L5) (**a**) and the proximal femur (**b**). (**c**) Bone strength in the lumbar spine (L2). (**d**) Serum osteocalcin levels. (**e**) Urinary CTX levels. (**f**) Bone formation in cancellous bone in the lumbar spine (L1), revealed by tetracycline labelling followed by calcein labelling 5 days later. Scale bar, 10 μm. (**g**) Serum Ca levels on day 84. Data are represented as the mean + s.e.m. of one experiment ($n = 9$ for Sham, $n = 12$ for vehicle and hPTH(1–34), $n = 10$ for 10 mg kg$^{-1}$ PCO371, $n = 11$ for 3 and 30 mg kg$^{-1}$ PCO371). Student's $t$-test was used to compare between the sham and OVX vehicle-treated groups; $\#P < 0.05$. Parametric Dunnett's test was used to compare PCO371- or hPTH(1–34)-treated groups with the OVX vehicle-treated group; $*P < 0.05$, $**P < 0.01$, $***P < 0.001$.

PCO371 is expected to provide three advantages: robust serum Ca control, low risk of hypercalciuria and a decrease in pill burden. However, we should bear in mind that rat skeleton is different from human skeleton, thus non-rodent animal models for

studying human bone metabolism are needed to bridge the gap between animal studies and clinical trials. We therefore examined the effects of PCO371 on normal dogs, since dogs have been used to study the human skeleton because of their extensive basic

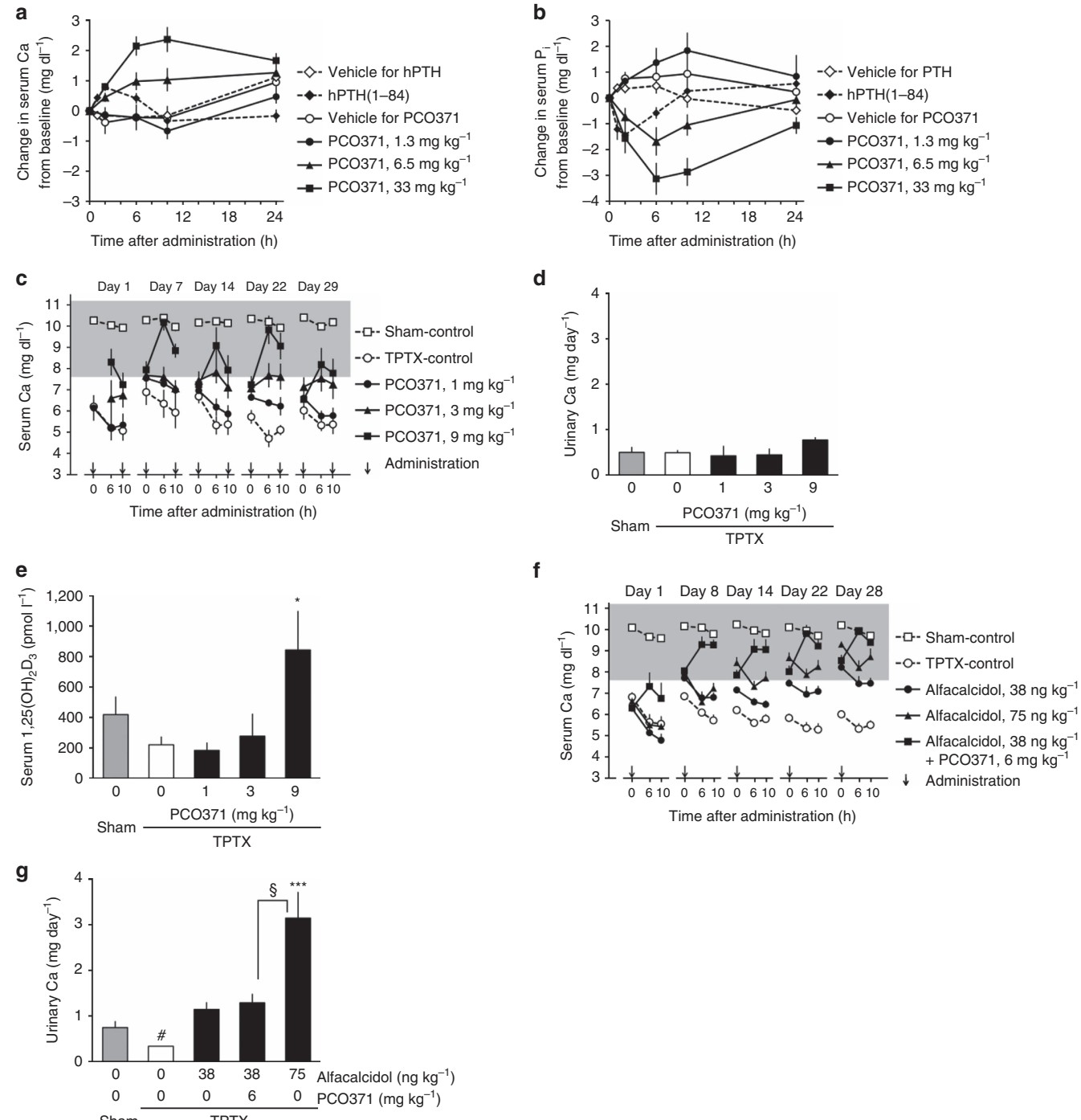

**Figure 6 | Effects of PCO371 on serum parameters in TPTX rats.** (**a,b**) Calcemic (**a**) and hypophosphatemic (**b**) effects of oral PCO371 or subcutaneous hPTH(1–84) in single administration. (**c–e**) Changes in serum Ca level (**c**), urinary Ca excretion (**d**), and serum 1,25(OH)$_2$D$_3$ level (**e**) from twice-daily repeated oral dosing of PCO371 (arrows in **c**) for 4 weeks. (**f,g**) Changes in serum Ca level (**f**), urinary Ca excretion (**g**) in once-daily repeated oral dosing of PCO371 (arrows) as an add-on to oral alfacalcidol treatment for 4 weeks. The shaded area shows the target therapeutic range (7.6–11.2 mg dl$^{-1}$) of serum Ca level. Data are represented as the mean + or ± s.e.m. of one experiment ($n = 6$ in **a,b,f** and **g**, $n = 5$ in **c–e**). Student's $t$ test was used to compare the sham and TPTX vehicle-treated groups (#$P < 0.05$ in **g**) and the alfacalcidol 75 ng ml$^{-1}$ and TPTX control groups (§$P < 0.05$ in **g**.) Parametric Dunnett's test was used to compare each treated group with the TPTX control group; *$P < 0.05$, ***$P < 0.001$ in **e** and **g**.

multicellular unit-based remodelling. In a pharmacological study with normal dogs, serum Ca was significantly increased by single oral administrations of PCO371 (3.0 − 30 mg kg$^{-1}$; Supplementary Fig. 5). Since no serious side effects of PCO371 were observed in preclinical toxicology studies, a phase 1 clinical study of PCO371 is currently being performed.

In the treatment of osteoporosis, in contrast to hypoparathyroidism, a sharp, transient increase in PTH levels is warranted to obtain a desirable anabolic effect on bone[28]. Once-daily injection of hPTH(1–34) giving a spiked pharmacokinetics profile is a widely applied osteoporosis therapy in clinic[16,31]. In the present study, orally administered PCO371 significantly increased bone

turnover, but induced limited increase in bone mass in OVX rats. In contrast, intravenous injection of PCO371 (10 mg kg[−1]) showed a spiked pharmacokinetics profile and significantly increased both BMD and bone strength to levels comparable with those produced by hPTH(1–34) (0.9 nmol kg[−1]), although markers of bone formation and resorption were more increased by PCO371 than they were by hPTH(1–34).

The results obtained in the present study, however, do not support the benefits of oral PCO371 as an anti-osteoporosis drug; further studies at different doses and intervals or studies in other animals, such as dog and monkey, are needed to better evaluate the potential of the compound for the treatment of osteoporosis.

Supra-therapeutic doses of hPTH(1–34) and hPTH(1–84) have been shown to increase the incidence of osteosarcoma in rat carcinogenicity studies. Because of this potential risk, hPTH(1–34) currently carries a black-box warning for the treatment of osteoporosis, although there appears to be no evidence of an increased risk of osteosarcoma in human subjects receiving the PTH peptide[16]. The exact mechanism is not yet elucidated, but it is postulated that profound bone formation in rats caused by PTH peptides is somewhat causative for osteosarcoma[32]. PCO371 behaves as a full agonist at hPTHR1 suggesting the possibility that the compound could increase the risk of osteosarcoma. Although profound bone formation similar to that caused by hPTHs is unlikely to be produced by oral PCO371, further studies are warranted to assess this potential safety concern.

Class B GPCRs serve as receptors and mediators of various endocrine peptide hormones, such as PTH, calcitonin and GLP-1 (refs 7,33). Although a few papers report the discovery of small-molecule agonists of class B GPCRs, to our knowledge none of them has been developed for clinical use, possibly due to insufficient oral bioavailability or to their characteristics as only partial agonists[17,18,34,35]. PCO371 could be the first clinical example, to our knowledge, of an orally active class B GPCR agonist.

A general mechanism of peptide binding for class B GPCRs has emerged, termed the two-domain model. In this mechanism, the C-terminal region of the ligand binds to the N-terminal extracellular domain of the receptor, and the N-terminal region of the ligand interacts with the transmembrane domain to activate the receptor and induce intracellular signalling[19,20,36]. PCO371 activated PTHR1 without interacting with the N-terminal extracellular domain of the receptor, suggesting that PCO371 activated PTHR1 solely by binding to the transmembrane domain of PTHR1, or more specifically, by interacting entirely or partially with TM6. We base this suggestion on the finding that Proline 415 within TM6 plays a critical role in the selective activation of PTHR1 by PCO371. The presence of a constitutively activating mutation (T410P) within TM6 (ref. 37) further supports our conclusion. Furthermore, the importance of TM6 in GPCR-mediated signalling has also been demonstrated by crystal structure analyses of corticotropin-releasing factor receptor 1 (ref. 38) and the β2 adrenergic receptor-Gs protein complex[39]. Studies involving the mechanism of action of PCO371 on PTHR1 will provide useful insights into the mechanism of activation of class B GPCRs.

In conclusion, the small-molecule PTHR1 agonist PCO371 is a compound with potential to be developed and used as an orally available drug to treat hypoparathyroidism. PCO371 is also useful to more fully understand the mechanism of activation of class B GPCRs.

## Methods

**Synthesis of PCO371.** Unless otherwise noted, all materials were obtained from commercial suppliers and used without further purification (Fig. 7). Silica gel

**Figure 7 | Synthesis of PCO371.** Reagents and conditions: (**a**) 4-trifluoromethoxybenzoic acid, HATU, DIPEA and DMF, room temperature; (**b**) 30% $H_2O_2$ aq., 2 mol l[−1] NaOH aq., EtOH, room temperature, then 5 mol l[−1] NaOH aq., 50 °C; (**c**) 4 mol l[−1] HCl in dioxane, $CH_2Cl_2$, room temperature; (**d**) 2-chloroethane-1-sulfonyl chloride, $Et_3N$, $CH_2Cl_2$, room temperature. (**e**) 1-(4-bromo-3,5-dimethylphenyl)-5,5-dimethylimidazolidine-2,4-dione, Pd(dba)₂, $t$-Bu₃P-HBF₄, methyl dicyclohexylamine, NMP, 100 °C; (**f**) 20% Pd(OH)₂/C, $H_2$, DMF–MeOH–MeCN, room temperature.

chromatography purification was performed using prepacked silica gel cartridges (Biotage, Shoko Scientific). Reverse-phase column chromatography purification was performed using Wakosil 25C18 (Wako Pure Chemical Industries). Nuclear magnetic resonance (NMR) spectra were determined with a Varian MR-400 spectrometer (400 MHz, Agilent).

Chemical shifts are shown in parts per million (p.p.m., $\delta$ units). The following NMR abbreviations are used: s = singlet, d = doublet, q = quartet, m = multiplet, dd = doublet of doublets, brs = broad singlet. High-resolution mass spectrometry (HRMS) was performed on a Xevo G2-S Tof instrument (Waters).

**Synthesis of compound 2.** To a stirred mixture of *tert*-butyl 4-amino-4-cyanopiperidine-1-carboxylate (**1**) (11.5 g, 50.9 mmol), 4-trifluoromethoxybenzoic acid (10 g, 48.5 mmol), and diisopropylethylamine (DIPEA) (8.78 g, 67.9 mmol) in anhydrous DMF (150 ml) was added 2-(3*H*-[1,2,3]triazolo[4,5-*b*]pyridin-3-yl)-1,1,3,3-tetramethylisouronium hexafluorophosphate(V) (HATU; 22.1 g, 58.2 mmol) at 0 °C. The mixture was stirred at room temperature overnight, quenched by 0.5 mol l$^{-1}$ HCl aq., and the product was extracted with EtOAc. The organic layer was washed with H$_2$O and sat. aq. NaCl, dried over anhydrous Na$_2$SO$_4$, and concentrated *in vacuo*. The crude solid was washed with EtOAc-hexane to afford *tert*-Butyl 4-Cyano-4-[4-(trifluoromethoxy)benzamido] piperidine-1-carboxylate (**2**) (18.9 g, 94%) as a white solid. $^1$H-NMR (400 MHz, CDCl$_3$) $\delta$: 7.84–7.81 (2H, m), 7.29 (2H, d, $J = 8.1$ Hz), 6.25 (1H, s), 4.14–3.97 (2H, m), 3.34–3.24 (2H, m), 2.61–2.47 (2H, m), 1.90–1.81 (2H, m), 1.47 (9H, s; Supplementary Fig. 6). $^{13}$C-NMR (100 MHz, CDCl$_3$) $\delta$: 165.54, 154.31, 152.15 (q, $J_{CF} = 1.9$ Hz), 131.39, 129.16, 120.79, 120.27 (q, $J_{CF} = 129.7$ Hz), 81.27, 80.57, 50.97, 40.00, 34.82, 28.37 (Supplementary Fig. 7). HRMS (electrospray ionization-time-of-flight (ESI-TOF)) *m/z* calcd for C$_{19}$H$_{21}$F$_3$N$_3$O$_4$ (M–H)$^-$, 412.1484; found 412.1493.

**Synthesis of compound 3.** To a stirred mixture of *tert*-butyl 4-cyano-4-[4-(trifluoromethoxy)benzamido]piperidine-1-carboxylate (**2**) (12.6 g, 30.5 mmol) in EtOH (150 ml) was added 2 mol l$^{-1}$ NaOH aq. (30.4 ml) followed by 30% H$_2$O$_2$ aq. (15.5 ml) at 0 °C. The mixture was stirred at room temperature for 1 h. Then 5 mol l$^{-1}$ NaOH aq. (30.5 ml) was added at room temperature. The mixture was stirred at 50 °C for 2 h. This mixture was concentrated *in vacuo* and EtOAc and NH$_4$Cl were added. The organic layer was washed with H$_2$O and sat. aq. NaCl and dried over anhydrous Na$_2$SO$_4$. The organic layer was concentrated *in vacuo* to afford *tert*-butyl 2-[4-(trifluoromethoxy)phenyl]-4-oxo-1,3,8-triazaspiro[4.5]dec-1-ene-8-carboxylate (**3**) (11.3 g, 90%) as a white solid. $^1$H-NMR (400 MHz, CDCl$_3$) $\delta$: 10.61 (1H, s), 8.02 (2H, d, $J = 8.9$ Hz), 7.37 (2H, d, $J = 8.9$ Hz), 4.18–3.96 (2H, m), 3.54–3.37 (2H, m), 2.05–1.89 (2H, m), 1.60–1.53 (2H, m), 1.51 (9H, sl; Supplementary Fig. 8). $^{13}$C-NMR (100 MHz, CDCl$_3$) $\delta$: 188.89, 156.52, 154.83, 151.83, 128.80, 126.97, 121.13, 120.34 (q, $J_{CF} = 258.6$ Hz), 79.78, 70.49, 39.81, 32.54, 28.50 (Supplementary Fig. 9). HRMS (ESI-TOF) *m/z* calcd for C$_{19}$H$_{21}$F$_3$N$_3$O$_4$ (M–H)$^-$, 412.1484; found 412.1502.

**Synthesis of compound 4.** A mixture of *tert*-butyl 2-[4-(trifluoromethoxy)phenyl]-4-oxo-1,3,8-triazaspiro[4.5]dec-1-ene-8-carboxylate (**3**) (11.3 g, 27.3 mmol) and 4 mol l$^{-1}$ HCl in dioxane (68.2 ml, 273 mmol) in CH$_2$Cl$_2$ (68 ml) was stirred at room temperature for 3 h. This mixture was concentrated *in vacuo*, and 1 mol l$^{-1}$ K$_3$PO$_4$ aq. was added. The product was extracted with EtOAc (300 ml)–EtOH (15 ml; twice). The organic layer was dried over anhydrous Na$_2$SO$_4$, and concentrated *in vacuo* to afford 2-[4-(trifluoromethoxy)phenyl]-1,3,8-triazaspiro [4.5]dec-1-en-4-one (**4**) (7.66 g, 90%) as a white solid. $^1$H-NMR (400 MHz, CD$_3$OD) $\delta$: 8.08 (2H, d, $J = 8.8$ Hz), 7.45 (2H, d, $J = 8.8$ Hz), 3.32–3.17 (4H, m), 2.05–1.95 (2H, m), 1.75–1.66 (2H, m; Supplementary Fig. 10). $^{13}$C-NMR (100 MHz, CD$_3$OD) $\delta$: 189.71, 153.33, 153.31, 130.81, 128.88, 122.19, 121.84 (q, $J_{CF} = 257.1$ Hz), 69.09, 42.00, 32.90 (Supplementary Fig. 11). HRMS (ESI-TOF) *m/z* calcd for C$_{14}$H$_{15}$F$_3$N$_3$O$_2$ (M + H)$^+$, 314.1116; found 314.1118.

**Synthesis of compound 5.** To a stirred mixture of 2-[4-(trifluoromethoxy)phenyl]-1,3,8-triazaspiro[4.5]dec-1-en-4-one (**4**) (6.56 g, 20.9 mmol) and Et$_3$N (11.1 ml, 83.8 mmol) in anhydrous CH$_2$Cl$_2$ (210 ml), 2-chloroethane-1-sulfonyl chloride (2.4 ml, 23 mmol) was added at 0 °C. The mixture was stirred at room temperature for 1 h under N$_2$. This mixture was quenched by H$_2$O, and the product was extracted with CH$_2$Cl$_2$. The organic layer was washed with H$_2$O and sat. aq. NaCl,

dried over anhydrous Na$_2$SO$_4$, and evaporated *in vacuo*. The crude residue was purified by column chromatography (silica gel, CH$_2$Cl$_2$–MeOH) to afford 2-[4-(trifluoromethoxy)phenyl]-8-(vinylsulfonyl)-1,3,8-triazaspiro[4.5]dec-1-en-4-one (**5**) (4.59 g, 54%) as a white solid. $^1$H-NMR (400 MHz, CDCl$_3$) $\delta$: 10.78 (1H, s), 8.02 (2H, d, $J = 8.8$ Hz), 7.38 (2H, d, $J = 8.8$ Hz), 6.54 (1H, dd, $J = 16.6$, 10.0 Hz), 6.31 (1H, d, $J = 16.6$ Hz), 6.09 (1H, d, $J = 10.0$ Hz), 3.81–3.74 (2H, m), 3.35–3.27 (2H, m), 2.23–2.14 (2H, m), 1.71–1.64 (2H, m; Supplementary Fig. 12). $^{13}$C-NMR (100 MHz, CDCl$_3$) $\delta$: 188.61, 156.96, 152.00, 132.99, 128.85, 128.49, 126.66, 121.15, 120.33 (q, $J_{CF} = 259.0$ Hz), 69.53, 41.81, 32.49 (Supplementary Fig. 13). HRMS (ESI-TOF) *m/z* calcd for C$_{16}$H$_{17}$F$_3$N$_3$O$_4$S (M + H)$^+$, 404.0892; found 404.0893.

**Synthesis of compound 6.** A mixture of 2-[4-(trifluoromethoxyl)phenyl]-8-(vinylsulfonyl)-1,3,8-triazaspiro[4.5]dec-1-en-4-one (**5**) (3.07 g, 7.60 mmol), 1-(4-bromo-3,5-dimethylphenyl)-5,5-dimethylimidazolidine-2,4-dione (**9**) (2.60 g, 8.36 mmol), Pd(dba)$_2$ (1.33 g, 2.31 mmol), methyl dicyclohexylamine (2.3 ml, 10.7 mmol), and tri-*tert*-butylphosphine tetrafluoroboric acid adduct (672 mg, 2.32 mmol) in anhydrous NMP (7.6 ml) was stirred at 100 °C for 2 h under N$_2$. The reaction mixture was diluted with EtOAc. The organic layer was washed with H$_2$O, dried over anhydrous MgSO$_4$ and evaporated *in vacuo*. The crude residue was purified by column chromatography (silica gel, EtOAc-hexane) to afford (*E*)-1-{3,5-dimethyl-4-[2-({4-oxo-2-[4-(trifluoromethoxy)lphenyl]-1,3,8-triazaspiro [4.5]dec-1-en-8-yl}sulfonyl)vinyl]phenyl}-5,5-dimethylimidazolidine-2,4-dione (**6**) (3.97 g, 82%) as a colourless foam. $^1$H-NMR (DMSO-d$_6$) $\delta$: 11.75 (1H, s), 11.17 (1H, s), 8.09 (2H, d, $J = 8.9$ Hz), 7.55 (2H, d, $J = 8.9$ Hz), 7.49 (1H, d, $J = 15.9$ Hz), 7.15 (2H, s), 7.01 (1H, d, $J = 15.9$ Hz), 3.67–3.59 (2H, m), 3.29–3.19 (2H, m), 2.39 (6H, s), 1.96–1.86 (2H, m), 1.70–1.62 (2H, m), 1.35 (6H, s; Supplementary Fig. 14). $^{13}$C-NMR (100 MHz, DMSO-d$_6$) $\delta$: 186.31, 177.15, 157.59, 154.55, 150.50, 139.82, 137.25, 134.58, 131.87, 129.07, 128.21, 128.10, 127.65, 121.11, 119.84 (q, $J_{CF} = 257.5$ Hz), 67.65, 63.96, 41.65, 31.93, 23.36, 20.48 (Supplementary Fig. 15). HRMS (ESI-TOF) *m/z* calcd for C$_{29}$H$_{31}$F$_3$N$_5$O$_6$S (M + H)$^+$, 634.1947; found 634.1969.

**Synthesis of PCO371.** To a stirred solution of (*E*)-1-{3,5-Dimethyl-4-[2-({4-oxo-2-[4-(trifluoromethoxyl)phenyl]-1,3,8-triazaspiro[4.5]dec-1-en-8-yl}sulfonyl)vinyl] phenyl}-5,5-dimethylimidazolidine-2,4-dione (**6**) (3.81 g, 6.01 mmol) in DMF (15 ml), MeOH (100 ml), and MeCN (100 ml) was added 20% Pd(OH)$_2$ on carbon (50% wet) (3.98 g, 2.83 mmol) under N$_2$, and this mixture was stirred at room temperature for 1 h under H$_2$. The mixture was filtered and concentrated *in vacuo*, and dissolved in EtOAc. The organic layer was washed with H$_2$O, dried over anhydrous MgSO$_4$, and evaporated *in vacuo*. The residue was purified by column chromatography (silica gel, EtOAc–hexane) to afford 1-{3,5-dimethyl-4-[2-({4-oxo-2-[4-(trifluoromethoxy)phenyl]-1,3,8-triazaspiro[4.5]dec-1-en-8-yl}sulfonyl) ethyl]phenyl}-5,5-dimethylimidazolidine-2,4-dione (**PCO371**) (3.03 g, 79%) as a white solid. $^1$H-NMR (400 MHz, CDCl$_3$) $\delta$: 9.37 (1H, brs), 7.93 (2H, d, $J = 8.0$ Hz), 7.73 (1H, brs), 7.35 (2H, d, $J = 8.0$ Hz), 6.94 (2H, s), 3.88–3.81 (2H, m), 3.53–3.45 (2H, m), 3.25–3.19 (2H, m), 3.07–3.00 (2H, m), 2.40 (6H, s), 2.19–2.10 (2H, m), 1.78–1.70 (2H, m), 1.47 (6H, s; Supplementary Fig. 16). $^{13}$C-NMR (100 MHz, CDCl$_3$) $\delta$: 187.73, 176.67, 156.89, 154.62, 151.95, 137.99, 135.80, 131.83, 128.98, 128.80, 126.63, 121.10, 120.30 (q, $J_{CF} = 258.6$ Hz), 69.27, 65.21, 47.43, 41.97, 32.72, 23.68, 23.00, 19.91 (Supplementary Fig. 17). HRMS (ESI-TOF) *m/z* calcd for C$_{29}$H$_{33}$F$_3$N$_5$O$_6$S (M + H)$^+$ 636.2103; found 636.2106.

**Synthesis of compound 9.** A mixture of 4-bromo-3,5-dimethylaniline (**7**) (3.47 g, 17.3 mmol), DIPEA (5.3 ml, 30.4 mmol) and 2-bromoisobutyric acid (3.86 g, 23.1 mmol) in anhydrous DMI (13 ml) was stirred at 100 °C for 1 h under N$_2$ (Fig. 8). Then additional 2-bromoisobutyric acid (496 mg, 2.97 mmol) and iPr$_2$EtN (0.80 ml, 4.59 mmol) was added, and the mixture was stirred at 100 °C for 1 h under N$_2$. The mixture was cooled to room temperature, and MeOH (52 ml) and 5 mol l$^{-1}$ NaOH aq. (52 ml, 260 mmol) were added. This mixture was stirred at 75 °C for 1.5 h and acidified with 1 mol l$^{-1}$ KHSO$_4$ aq. (pH 5) at 0 °C, diluted with H$_2$O, and the product was extracted with EtOAc. The organic layer was washed with H$_2$O, dried over anhydrous MgSO$_4$ and evaporated *in vacuo* to afford compound **8** (5.79 g, crude). This crude mixture was used in the next step without further purification.

To a stirred solution of the above crude mixture (5.79 g) in CH$_2$Cl$_2$ (62 ml) and AcOH (62 ml), NaOCN (5.03 g, 77.4 mmol) was added at room temperature. The mixture was stirred at room temperature for 3 h and poured into sat. aq. NaHCO$_3$ (400 ml). The reaction mixture was basified with 5 mol l$^{-1}$ NaOH aq. (pH 7–8),

**Figure 8 | Synthesis of compound 9.** Reagents and conditions: (**g**) 2-bromoisobutyric acid, DIPEA, DMI, 100 °C; (**h**) 5 mol l$^{-1}$ NaOH aq., MeOH, 75 °C; (**i**) NaOCN, AcOH, CH$_2$Cl$_2$, room temperature.

and the product was extracted with EtOAc. The organic layer was dried over anhydrous $MgSO_4$ and evaporated *in vacuo*. The crude solid was washed with EtOAc-hexane and $CH_2Cl_2$-hexane, consecutively, and dried to afford 1-(4-bromo-3,5-dimethylphenyl)-5,5-dimethylimidazolidine-2,4-dione (**9**) (3.80 g, 71% from compound **7**). $^{1}$H-NMR (400 MHz, CDCl$_3$) $\delta$: 8.64 (1H, s), 6.96 (2H, s), 2.44 (6H, s), 1.45 (6H, s; Supplementary Fig. 18). $^{13}$C-NMR (100 MHz, CDCl$_3$) $\delta$: 176.50, 154.57, 139.91, 131.92, 128.47, 128.12, 65.11, 24.03, 23.68. (Supplementary Fig. 19) HRMS (ESI-TOF) *m/z* calcd for $C_{13}H_{14}BrN_2O_2$ (M-H)$^{-}$, 309.0239; found 309.0240

**Screening and optimization.** To identify small-molecule agonists of PTHR1, we used a cell-based high-throughput screening method employing LLC-PK1 cells (ATCC) expressing hPTHR1 (HKRK-B7 cells) and using urokinase-type plasminogen activator expression as a readout[40]. The hit compounds found from our compound library were then evaluated for their ability to increase cAMP in HKRK-B7 cells. Parental LLC-PK1 cells, which express porcine calcitonin receptors but not hPTHR1, were used as a negative control. The hit compounds were optimized for potency and other properties such as solubility and metabolic stability to identify a lead compound possessing *in vitro* and *in vivo* activities similar to that of hPTH(1–34). This compound was further optimized to improve oral bioavailability to yield a clinical candidate compound, PCO371.

**cAMP production assay.** COS-7 cells (ATCC) were seeded into a 96-well flat-bottomed plate ($1.4 \times 10^4$ cells per 100 μl per well) and incubated overnight. The next day, the COS-7 cells were transfected by adding 10 μl of a mixture of hPTHR1, hPTHR2 or mutant hPTHR1 plasmid, rat PTHR1 or rat PTHR2 plasmid, or empty vector (0.1 μg per well), FuGENE HD Transfection Reagent (0.4 μl per well, Promega), and Opti-MEM I Reduced Serum Medium (10 μl per well, Life Technologies) to each well. On the third day after cell seeding, the culture medium in the wells was discarded, and 50 μl of PCO371 free base and hPTH(1–34) (Peptide Institute, Inc) serially diluted in assay medium (2 mg ml$^{-1}$ BSA, 1 mmol l$^{-1}$ 3-isobutyl-1-methylxanthine (IBMX), and 2 mmol l$^{-1}$ HEPES/McCoy's 5A medium, Life Technologies) was added to each well. The plate was placed in a 37 °C constant-temperature water bath for 20 min to allow the reaction to proceed. After incubation, the liquid in the wells was discarded, the wells were washed with the assay medium (100 μl per well), and the plate was frozen on dry ice. Next, 40 μl of 50 mmol l$^{-1}$ HCl was added to each well, and the plate was stored in a freezer set at $\leq -20$ °C. The cAMP concentrations were measured by using a cAMP EIA kit (GE Healthcare). EC$_{50}$ values were determined from three independent experiments in duplicate by a non-linear regression E$_{max}$ model (JMP ver.9.0.2, SAS Institute). The geometric mean of the EC$_{50}$ values obtained in three experiments in duplicate was calculated using Microsoft Office Excel 2007. UMR-106 cells (ATCC) were cultured for 1 day, and the cells were treated with PCO371 (10 nmol l$^{-1}$ to 0.1 mmol l$^{-1}$) or hPTH(1–34) (0.1 pmol l$^{-1}$ to 9 μmol l$^{-1}$) in the same assay medium as that used for the COS-7 cells for 20 min at 37 °C, and incubations were terminated with 50 mmol l$^{-1}$. The cAMP concentrations were determined using a cAMP EIA kit (GE Healthcare).

**Phospholipase C activity assay.** COS-7 cells (ATCC) were seeded into a 96-well flat-bottomed plate ($1.4 \times 10^4$ cells per 100 μl per well) and incubated overnight. The next day, the COS-7 cells were transfected by adding 10 μl of a mixture of hPTHR1 plasmid or empty vector (0.1 μg per well), FuGENE HD Transfection Reagent (0.4 μl per well, Promega), and Opti-MEM I Reduced Serum Medium (10 μl per well, Life Technologies) to each well. On the second day after the cells were seeded into the plate, the culture medium in the wells was discarded, and 100 μl of $^{3}$H-myoinositol/0.1% BSA/DMEM adjusted to 111 kBq ml$^{-1}$ (PerkinElmer, Inc.) was added to each well. On the third day after the cells were seeded into the plate, the culture medium in the wells was discarded, each well was washed with 100 μl of binding buffer ( + LiCl; 50 mmol l$^{-1}$ Tris-HCl, 100 mmol l$^{-1}$ NaCl, 5 mmol l$^{-1}$ KCl, 2 mmol l$^{-1}$ CaCl$_2$, 5% horse serum, 0.5% FBS, and 30 mmol l$^{-1}$ LiCl), and 50 μl of PCO371 free base (0.1 μmol l$^{-1}$ to 0.1 mmol l$^{-1}$) or hPTH(1–34) (10 pmol l$^{-1}$ to 0.1 mmol l$^{-1}$) serially diluted with binding buffer ( + LiCl) was added to each well. The plate was placed in a CO$_2$ incubator set at 37 °C for 40 min to allow the reaction to proceed. After incubation, the liquid in the wells was discarded, and each well was washed with 100 μl of binding buffer ( + LiCl). A 100-μl aliquot of 5% trichloroacetic acid (TCA) solution was added to each well, and the plate was left to stand for about 3 h on ice. The TCA solution in each well was recovered into a microtube, and then another 100 μl of 5% TCA solution was added to each well and recovered into the same microtube. Next, 400 μl water-saturated ether was added to each tube, the contents of the tube were thoroughly mixed, and the ether was aspirated. These three steps were repeated, and then 20 μl Tris-NaOH was added to each tube. A POLY PREP chromatography column (Bio-Rad Laboratories, Inc.) was packed with AG1-X8 Resin (Bio-Rad Laboratories, Inc.) and washed with 10 ml of 10 mmol l$^{-1}$ myoinositol, and then the above sample was added. The column was washed with 10 ml of 10 mmol l$^{-1}$ myoinositol, and then lipids were washed out by adding 8 ml of 5 mmol l$^{-1}$ Borax/60 mmol l$^{-1}$ NH$_4$COOH. Inositol phosphates were eluted using 3 ml of 1 mol l$^{-1}$ NH$_4$COOH/100 mmol l$^{-1}$ HCOOH, and then 10 ml of ULTIMA-FLO AF (PerkinElmer, Inc.) was added. After that, the $^{3}$H radioactivity was measured using a liquid scintillation analyzer. EC$_{50}$ values were determined

from three independent experiments in triplicate by a non-linear regression E$_{max}$ model (JMP ver.9.0.2, SAS Institute). In addition, EC$_{50}$ of PCO371 free base was defined as the concentration of PCO371 free base that achieves the EC$_{50}$ for hPTH(1–34). The geometric means of the EC$_{50}$ values obtained in three experiments in duplicate were calculated using Microsoft Office Excel 2007.

**Mutant receptor construction.** Mutant hPTHR1 (hPTHR1-delNT) was constructed by removing N-terminal residues 23–181 and inserting a nine-amino acid influenza hemagglutinin (HA) tag joined to a tetraglycine linker (YPYDVPDYA-GGGG-) between Ala-22 and Glu-182.

Wild-type human or mutant PTHR1s (C1, C2, C3, M24, M25, M26, M414V, P415L and L416V) were generated by PCR-mutagenesis, and the sequences were confirmed by replacement of non-functional N-terminal residues 93–101 (EDKEAPTGS) in the E2 region with a nine-amino acid HA tag (YPYDVPDYA) for HA binding. We confirmed that the cell-surface expression of mutant receptors on transfected COS-7 cells was comparable to that of HA-tagged wild-type hPTHR1 by means of an HA antibody binding assay using Anti-HA rabbit antibody (Sigma-Aldrich) and secondary $^{125}$I-labelled goat anti-rabbit IgG antibody (PerkinElmer) or by Cell-ELISA using Anti-HA mouse antibody (12CA5; Sigma-Aldrich) and Goat Anti-Mouse IgG HRP Secondary Antibody (Abgent).

**Receptor binding assay.** COS-7 cells (ATCC) were seeded into 10-cm dishes ($1.7 \times 10^6$ cells per dish) and incubated overnight. The next day, the COS-7 cells were transfected by adding 500 μl of a mixture of hPTHR1 plasmid (10 μg per dish), FuGENE HD Transfection Reagent (40 μl per dish, Promega) and Opti-MEM I Reduced Serum Medium (450 μl per well, Life Technologies) to each dish. On the third day after the cells were seeded into the dishes, the culture medium in the dishes was discarded, each dish was washed with 5 ml of Buffer A (10 mmol l$^{-1}$ Tris-HCl, pH 7.4/4 mmol l$^{-1}$ EDTA) without protease inhibitor cocktail (PI; Nacalai Tesque, Inc.), and then 750 μl of Buffer A (10 mmol l$^{-1}$ Tris-HCl, pH 7.4/4 mmol l$^{-1}$ EDTA/1 × PI) was added to each dish. The cells were recovered, and then homogenized on ice. The recovered cells were centrifuged (400 $g$, 4 °C, 10 min), and the supernatant was recovered. The supernatant was centrifuged again (15,300 $g$, 4 °C, 30 min), and the supernatant was removed leaving the membrane fraction, which was suspended in membrane buffer ( − BSA; 20 mmol l$^{-1}$ HEPES/0.1 mol l$^{-1}$ NaCl/3 mmol l$^{-1}$ MgSO$_4$/20% Glycerol/1 × PI). The protein concentration was determined using Protein Assay Dye Reagent Concentrate (Bio-Rad Laboratories, Inc). The membrane protein solution was stored in a freezer set at $\leq -80$ °C.

The binding of hPTH(1–34) and PCO371 free base to hPTHR1 was evaluated by competition assay using $^{125}$I-[Aib$^{1,3}$,M]PTH(1–15) as the competing entity. Membrane protein solution was prepared by dilution with membrane buffer ( + PI). PCO371 free base solution (0.1 μmol l$^{-1}$ to 0.1 mol l$^{-1}$) or hPTH(1–34) solution (0.1 pmol l$^{-1}$ to 1 μmol l$^{-1}$) serially diluted with membrane buffer ( + PI; 180 μl), membrane protein solution (10 μl) and tracer (10 μl) was added to a Multiscreen HTS 96 HV Opaque plate. After reaction for 1.5 h at room temperature, the reaction liquid was aspirated and the filter was washed with 200 μl of membrane buffer. The dried filter was recovered, and the $^{125}$I radioactivity was measured using a gamma counter. The experiment was performed independently three times in triplicate. Total $^{125}$I count rates (cpm) measured by the gamma counter were used as the primary data. Binding to hPTHR1 was evaluated by calculating the per cent specific binding from the following formula using Microsoft Office Excel 2007.

Percentage specific binding was calculated by the following formula: equation (1)

$$\% \text{ Specific binding} = [(\text{count rate in each sample} - \text{NSB})/B_0] \times 100 \quad (1)$$

The mean count rate for the samples with 1 μmol l$^{-1}$ hPTH(1–34) added was entered as the nonspecific binding (NSB), the mean count rate for unspiked samples was entered as the total count, and the total count minus the NSB was entered as B$_0$.

**Gene expression assay.** UMR-106 cells (ATCC) were incubated with DMEM with 0.2% BSA for 4 h and were then treated with hPTH(1–34) or PCO371 for 1 or 6 h. At 1 or 6 h after adding test compounds, total RNA was isolated by using an RNeasy Plus Kit (Qiagen). cDNA was transcribed from the isolated total RNA by using TaqMan Reverse Transcription Reagents (Thermo Fisher Scientific). mRNA levels were quantified by a real-time PCR system (Thermo Fisher Scientific). The relative quantity of transcripts was calculated by using the comparative threshold cycle (Ct) method and normalized to 18S ribosomal RNA as an internal control. The following cDNA-specific primers and probes were used: Fos (encoded by *Fos*, Rn02396759_m1), osteocalcin (encoded by *Bglap*, Rn00566386_g1), receptor activator of nuclear factor-κB ligand (RANKL, encoded by *Tnfsf11*, Rn00589229_m1), osteoprotegerin (osteoprotegerin, encoded by *Tnfrsf11b*, Rn00563499_m1) and sclerostin (encoded by *Sost*, Rn00577971_m1).

**cAMP washout assay.** The duration of cAMP signalling capacity of PCO371, hPTH(1–34) and LA-PTH (synthesized at American Peptide Company, Inc.) was

assessed in a time-course experiment in UMR-106 cells (cAMP washout assay). The cells were treated with assay medium ($2 \, mg \, ml^{-1}$ BSA, and $2 \, mmol \, l^{-1}$ HEPES/McCoy's 5A medium, Life Technologies) alone (basal) or with ligands ($1 \, \mu mol \, l^{-1}$ for hPTH(1–34), $0.1 \, \mu mol \, l^{-1}$ for LA-PTH and $0.1 \, mmol \, l^{-1}$ for PCO371) in the assay medium for 10 min, washed ($t = 0$), incubated for 0, 30, 60 and 120 min (washout phase), and then further incubated for 5 min in the assay medium containing IBMX ($2 \, mmol \, l^{-1}$). Incubations were terminated with $50 \, mmol \, l^{-1}$ HCl. The cAMP concentrations were determined using a cAMP EIA kit (GE Healthcare).

**Selectivity assay.** Selectivity assays against class B GPCRs were performed by Cerep (France). Agonistic and antagonistic effects of PCO371 against a panel of 12 class B GPCRs (CGRP, calcitonin, CRF1, CRF2α, GLP-1, GLP-2, glucagon, secretin, GHRH, PAC1, VPAC1 and VPAC2) were examined at PCO371 concentrations of 1 and $10 \, \mu mol \, l^{-1}$. The following cell lines were used; T47D (calcitonin), βTC6 (GLP-1), HT-29 (VPAC1, VPAC2), CHO (CGRP, CRF1, GLP-2, Glucagon, Secretin, GHRH and PAC1, recombinant), and HEK-293 (CRF-2, recombinant). Their antagonistic activity was measured in the presence of the control agonists shown in Supplementary Table 1. In agonist and antagonist experiments, intracellular cAMP levels were determined with the HTRF detection system. $EC_{50}$ and $IC_{50}$ values of control agonists or antagonists were calculated by non-linear regression analysis of the concentration-response curves generated with mean replicate values using Hill equation curve-fitting using the formula: equation (2).

$$Y = D + [(A - D)/(1 + (C/C_{50}) \times nH]$$ (2)

$Y = $ response, $A = $ left asymptote of the curve, $D = $ right asymptote of the curve, $C = $ compound concentration, $C_{50} = EC_{50}$ or $IC_{50}$, $nH = $ slope factor. Agonistic activity of PCO371 at 1 or $10 \, \mu mol \, l^{-1}$ was calculated as a per cent of the control agonist response; intracellular cAMP accumulation by PCO371 × 100/intracellular cAMP accumulation by the control agonist at the $EC_{50}$ concentration. Antagonistic activity was expressed as a per cent of the control agonist inhibition response (equation (3)).

$$100 - (\text{intracellular cAMP accumulation by PCO371}$$
$$\times 100/\text{intracellular cAMP accumulation by the control antagonist at the}$$
$$IC_{50} \text{ concentration})$$
(3)

A value $> 50\%$ was defined as positive (agonist or antagonist activity), a value between 25 and 50% as weak to moderate, and a value $< 25\%$ was considered as negative.

**Animal ethics.** In accordance with the Guidelines for the Care and Use of Laboratory Animals at Chugai Pharmaceutical, all animal studies were performed under the approval of the company's Institutional Animal Care and Use Committee. The company is fully accredited by the Association for Assessment and Accreditation of Laboratory Animal Care International (http://www.aaalac.org). The studies were also carried out in compliance with the 'Act on Welfare and Management of Animals' in Japan.

**Fetal rat long bone cultures.** Bone-resorbing activity was assessed by fetal rat long bone cultures[41]. $^{45}CaCl_2$ (1,480 kBq, PerkinElmer) was subcutaneously injected into a pregnant Crl:CD(SD) rat (Charles River Laboratories) on its 18th day of gestation. At 48 h after the injection, the pregnant rat was euthanized, and the ulnae and radial bones were isolated from euthanized fetal rats. The long bones were placed on nylon mesh and covered with paraffin paper, and a floating culture in BGJb medium (Life Technologies) containing antibiotic–antimycotic ($\times 1$, Life Technologies) was carried out for 3 days at 37 °C in 95% air and 5% $CO_2$ atmosphere. On the 4th day, the medium was replaced with fresh BGJb medium containing hPTH(1–34) ($1 \, nmol \, l^{-1}$ to $1 \, \mu mol \, l^{-1}$) or PCO371 ($0.1 \, nmol \, l^{-1}$ to $3 \, \mu mol \, l^{-1}$) or vehicle (0.1% DMSO and $0.1 \, mmol \, l^{-1}$ acetic acid at final concentration) and cultured for a further 4 days at 37 °C. Bone samples were then treated with 10% TCA (500 μl/bone) overnight at 37 °C. Radioactivity of the bone and the medium samples was measured[41]. Four to five long bones were used for each individual dose. $^{45}Ca$ release (%) was calculated by the following formula (equation (4)):

$$^{45}Ca \text{ release } (\%) = \text{medium radioactivity (cpm)} \times 100/$$
$$[\text{medium radioactivity (cpm)} + \text{bone radioactivity (cpm)}]$$
(4)

**Repeated administration study in OVX rats.** OVX was carried out in 32-week-old female Crl:CD (SD) rats (Charles River Laboratories). Twelve weeks after surgery, lumbar vertebral BMD (L3–L5) was measured *in vivo* by dual-energy X-ray absorptiometry (DXA) using a DCS-600EX-IIIR bone densitometer (Hitachi Aloka Medical), and the rats were divided into five groups, each with a similar mean value of BMD and body weight. Sample sizes were determined based on

previous experiments with hPTHs and a published study with ibandronate and eldecalcitriol in our laboratories[42] ($n = 7$–10). Then, OVX rats were treated once daily intravenously (vehicle or PCO371 at 3 or $10 \, mg \, kg^{-1}$) or orally (vehicle or PCO371 at $30 \, mg \, kg^{-1}$) or subcutaneously [vehicle or hPTH(1–34) at $0.9 \, nmol \, kg^{-1}$] for 12 weeks. Ten to twelve OVX rats were assigned to each dose group. Nine rats were assigned to the sham-operated control group and were treated with vehicle for PCO371. Ten per cent DMSO (Wako Pure Chemical Industries)/10% Kolliphor EL (Sigma-Aldrich) in 10% hydroxypropyl-β-cyclodextrin (HPCD; Nihon Shokuhin Kako)/0.752% glycine (Wako Pure Chemical Industries) buffer was used as vehicle for PCO371, and phosphate-citrate buffer (pH 6.0) was used as vehicle for hPTH(1–34). To label bone-forming surfaces, tetracycline ($25 \, mg \, ml^{-1}$, Nippon Zenyaku Kogyo) was subcutaneously injected on Day 78 and calcein ($8 \, mg \, kg^{-1}$, Dōjindo Laboratories) was subcutaneously injected on day 83. Under isoflurane anaesthesia, jugular vein blood was collected immediately before and at 1, 2, 6, 10 and 24 h after administration on day 82 to measure plasma concentration of PCO371 or on day 84 to measure serum Ca levels (o-CPC method, Wako Pure Chemical Industries). Urine was collected for 24 h from metabolic cages on days 84–85 to measure urinary collagen type 1 cross-linked C-telopeptide (CTX) by using the RatLaps ELISA System (Immunodiagnostic Systems). Under isoflurane anaesthesia, blood was collected on day 85 from the abdominal aorta to measure serum osteocalcin level by using the Rat Osteocalcin ELISA System (GE Healthcare). The lumbar spine and bilateral femurs were excised after 12 weeks of treatment. The first lumbar vertebra (L1), 3rd to 5th lumbar vertebrae (L3–L5), the right femur, and the right tibia were preserved in 70% ethanol to measure bone histomorphometry and BMD (DCS-600EX; Hitachi Aloka Medical). Measurement of bone histomorphometry was performed by the Ito Bone Histomorphometry Institute (Niigata, Japan). The second lumbar vertebra (L2) was wrapped in saline-soaked gauze and stored at $-80$ °C until measurement. Ultimate compressive strength (N) of the L2 vertebra was measured by using a mechanical testing machine (TK-252C; Muromachi Kikai).

**Single administration study in TPTX rats.** TPTX was carried out in 6-week-old female Crl:CD(SD) rats (Charles River Laboratories). Rats with serum Ca levels of $< 8.0 \, mg \, dl^{-1}$ at 5 days after surgery were used for the experiment. The rats were divided into six groups, each with a similar mean value of serum Ca levels and body weight. TPTX rats were treated once orally (vehicle or PCO371 at 1.3, 6.5 or $33 \, mg \, kg^{-1}$) or subcutaneously (vehicle or hPTH(1–84) (Chugai Pharmaceutical) at $9 \, nmol \, kg^{-1}$). Six TPTX rats were assigned to each dose group. The sample size was determined based on previous experiments and a published study with hPTHs and LA-PTH in our laboratories[24] ($n = 5$–6). Vehicle for PCO371 was the same as in the OVX rat study, and phosphate-citrate buffer (pH 6.0) was used for hPTH(1–84). Under isoflurane anaesthesia, jugular vein blood was collected immediately before and at 1, 2, 6, 10 and 24 h after administrations to measure serum Ca (o-CPC method, Wako Pure Chemical Industries) and $P_i$ levels (xanthine oxidase method, Wako Pure Chemical Industries).

In another experiment, the effects of a single treatment with PCO371 or hPTH(1–34) on serum Ca and $P_i$ in TPTX rats (6-week-old) were evaluated at the dose of 3, 9.5 or $30 \, mg \, kg^{-1}$ (PCO371, p.o.) or $9 \, nmol \, kg^{-1}$ (hPTH(1–34), s.c.) according to the same method as that described above.

**Repeated administration study in TPTX rats.** TPTX was carried out in 6-week-old female Crl:CD (SD) rats (Charles River Laboratories). Rats with serum Ca levels of $< 8.0 \, mg \, dl^{-1}$ at 5 days after surgery were used in the following experiments. The rats were divided into four groups both in study 1 and 2, each with a similar mean value of serum Ca levels and body weight. Sample sizes were determined as described in the above single administration study. To evaluate the calcemic effects of PCO371 and hPTHs, we defined a target therapeutic range for serum Ca of $7.6$–$11.2 \, mg \, dl^{-1}$, which corresponds to the low normal to normal range of serum Ca levels[10,11].

**Study 1.** TPTX rats were treated twice daily (at around 09:00 and 19:00 hours) orally (vehicle or PCO371 at 1, 3 or $9 \, mg \, kg^{-1}$) or subcutaneously [hPTH(1–34) or vehicle at $9 \, nmol \, kg^{-1}$] for 4 weeks. Four rats were assigned to sham control, and treated with vehicle for PCO371. Vehicles for PCO371 and hPTH(1–34) were the same as in the OVX rat study. Under isoflurane anaesthesia, jugular vein blood was collected immediately before and at 6 and 10 h (for PCO371) or at 1 and 10 h (for hPTH) after administrations to measure the serum Ca levels on days 1 (the first dosing day), 7, 14, 22 and 29. Urine was collected for 24 h by using metabolic cages on days 29–30, and Ca excretion was calculated from urine volume and Ca level. Under isoflurane anaesthesia, jugular vein blood was collected 10 h after administrations to measure serum $1,25(OH)_2D_3$ levels on Day 29 by using a $1,25(OH)_2D_3$ RIA kit (Fujirebio). Four or five TPTX rats were assigned to each dose group. Serum and urine Ca levels were measured in the same way as in the single-dosing study. In another experiment, TPTX rats were treated subcutaneously once daily with hPTH(1–84) at $9 \, nmol \, kg^{-1}$, for 4 weeks. Serum Ca was measured immediately before and at 2, 6 and 10 h after administration on day 1 (the first dosing day), 7, 14, 21 and 28 using the same method as described above. Urine was

collected for 24 h by using metabolic cages on days 28 and 29, and Ca excretion was calculated from urine volume and Ca level.

**Study 2.** TPTX rats were orally treated once daily (vehicle; alfacalcidol (Chugai Pharma Manufacturing) at 38 or 75 ng kg$^{-1}$; or alfacalcidol at 38 ng kg$^{-1}$ plus PCO371 at 6 mg kg$^{-1}$) for 4 weeks. Five or six TPTX rats were assigned to each dose group. The vehicle for alfacalcidol was medium-chain triglyceride (Chugai Pharma Manufacturing). Under isoflurane anaesthesia, jugular vein blood was collected immediately before and at 6 and 10 h after administration to measure serum Ca levels on day 1 (the first dosing day), 8, 14, 22 and 28. Urine was collected for 24 h by using metabolic cages on days 28 and 29. Serum and urine Ca levels were measured in the same way as in the single-dosing study.

**Single administration study in normal dogs.** Marshall Beagle dogs (20 months old, male, Marshall BioResources Japan Inc) were treated once orally (vehicle or PCO371 at 3, 10 or 30 mg kg$^{-1}$). Six dogs were assigned to each group. Vehicle for PCO371 was the same as in the rat studies. The sample size was determined based on previous experiments with hPTHs in our laboratories ($n = 6$). Blood was collected from the cephalic vein immediately before and at 1, 2, 4, 6, 8, 10 and 24 h after administration to measure serum Ca levels (o-CPC method, Wako Pure Chemical Industries).

**Pharmacokinetics studies.** PCO371 was administered at a dose of 2, 6 or 18 mg kg$^{-1}$ orally to 7-week-old normal female rats (RccHan: WIST, JLA; three rats for each dose group), or at a dose of 2, 6 or 18 mg kg$^{-1}$ orally to 7-week-old female TPTX rats [Crl:CD(SD), Charles River Laboratories; three TPTX rats for each dose group] or at 30 mg kg$^{-1}$ orally or 3, 10 mg kg$^{-1}$ intravenously to 55-week-old OVX rats as described above. Blood samples were collected at 15 and 30 min, and 1, 2, 4, 8 and 24 h after administration. Blood samples were also collected 2 min after intravenous administration in mature OVX rats. The concentrations of plasma PCO371 were determined by liquid chromatography–tandem mass spectrometry [LC-MS/MS (API-3200 (AB SCIEX), detection limit: 1 ng ml$^{-1}$). Pharmacokinetics parameters ($T_{1/2}$, $T_{max}$, $C_{max}$, AUC$_{inf}$ and AUC$_{0-24h}$) were calculated by a non-compartmental model using WinNonlin 6.4 (Pharsight). Oral bioavailability (BA) was calculated with AUC$_{inf}$ after oral and intravenous administration at a dose of 2 mg kg$^{-1}$ by using the following equation (5):

$$BA\ (\%) = AUC_{inf}, po/AUC_{inf}, iv \times 100 \tag{5}$$

hPTH(1–34) was subcutaneously injected at a dose of 3 nmol kg$^{-1}$ to normal female 10-week-old Crl:CD(SD) rats (Charles River Laboratories). Four rats were assigned to each group. Blood samples were collected at pre-dose and at 2, 5, 10, 15, 30 and 45 min, and 1 and 2 h after administration. The concentrations of plasma hPTH(1–34) were determined by an enzyme immuno assay (EIA) system (Immutopics International).

hPTH(1–84) was subcutaneously injected at a dose of 30, 100 or 300 nmol kg$^{-1}$ in TPTX female 8-week-old Crl:CD(SD) rats (Charles River Laboratories). Blood samples were collected from the jugular vein at pre-dose and at 5, 10, 15, 20, 30 and 45 min and at 1, 2, 4 and 24 h after administration. Five TPTX rats were assigned to each dose group. The concentrations of plasma hPTH(1–84) were determined by an EIA system (Immutopics International).

**Statistical analysis.** Statistical significance was assumed at the two-sided 5% level ($P < 0.05$). The data displayed normal variance. The dose-dependent effect was tested with Williams's test. Dunnett's test was performed to determine which data differed from the control. Student's $t$ test was performed to test for significant difference between two groups. SAS Preclinical Package (SAS Institute) software was used for statistical analyses for all other experiments. Quantitative results are represented as the mean ± s.d. or s.e.m., unless otherwise noted.

**Data availability.** The data that support the findings of this study are available from the corresponding author on request.

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

## Acknowledgements

We express our deep appreciation to the late E. Ogata, who emphasized to us the great need for improved therapy for hypoparathyroidism and wisely and helpfully encouraged us in our efforts to produce the compound reported here. We are grateful to H. Okabe for thoughtful discussion and helpful comments on the manuscript. We thank H. Saito, N. Inomata, H. Takasu, N. Shimizu and F. Makishima for their contribution to screening assays and hit identification. We also thank our colleagues at Chugai Medical Science Inc. for their support with the *in vitro* and *in vivo* experiments.

## Author contributions

T.T. and H.N. led the pharmacological research. T.E. and Y.N. led the medicinal chemistry studies. E.J., M.H., T.W., M.K., M.S. and H.K. conducted the pharmacological studies. K.O., T.M. and S.A. conducted the pharmacokinetics studies. Y.K. and H.S. directed this programme based on their biological and chemical expertise, respectively. T.T., H.N., M.S., H.K., T.E., Y.N. and Y.K. wrote the manuscript. T.T., M.S., H.K., T.E. and Y.N. revised the manuscript.
