## [Peer review file · Nature Communications]

Reviewers' comments:

Reviewer #1 (Expert in GPCRs; Remarks to the Author):

General and major comments

The type 1 PTH-receptor (PTHr) is a GPCR regulating calcium and phosphate homeostasis, and so far the only therapeutic target to stimulate bone anabolism when PTH(1-34) is daily administered via subcutaneous injection. The PTHr is thus a physiologically and medically important receptor under intense basic and clinical research activities. The new study of Tamura and colleagues now reports the identification of a new small molecule called PCO371 selective for the PTHr, which acts as a long-acting PTH analog as it relates to hypercalcemic (elevated serum Ca) and hypophosphatemic responses when administered orally or subcutaneously to rats. The PCO371 compound has therefore the potential to become a new and more efficient therapy for the treatment of hypoparathyroidism (hypocalcemia).

Although the new *in vivo* observations made with PCO371 are important on their own, the major issue of this manuscript is the total absence of a mechanism by which PCO371 acts on the PTHr to stimulate prolonged calcemic and phosphatemic responses. This is particularly critical given that recent studies (not discussed in this manuscript) showed that certain peptidic PTH analogs (known as LA-PTH) trigger similar Ca and P responses, but also display high affinity for PTHr (in its G-protein coupled or uncoupled states) and stimulate sustained cAMP from internalized LAPTH-bound PTHr complexes (See studies from Gardella and Vilardaga's labs in PNAS 2008, PNAS 2013, NIH, and reviewed in Nat Chem Biol 2014 and Pharm Rev 2015). The 10,000-fold reduced affinity of PCO371 for PTHr (Fig. 1) presumably reflects a very rapid dissociation of the ligand and thus little residence time on the receptor and short-lived cAMP signaling. If this prediction is proved as correct, it would be difficult to reconcile binding and signaling properties of LA-PTH with those of PCO371. The authors should address this issue by additional experiments before this manuscript becomes a good candidate for Nat Comm.

Minor comments

1. Experiments with UMR106: It is unclear how 1 μ M of PCO371 is unable to produce cAMP (Suppl. Fig. 2C) and IP, but still stimulates significant stimulation of Fos, BGLAP, RANKL transcripts and inhibition of OPG and SOST (Fig. 2).
2. The authors should note that cAMP data done with COS cells (Fig. 1) do not demonstrate that PCO371 is a full agonist due to the unavoidable presence of spare receptors when cells are transiently transfected. However, data with UMR106 cells expressing low levels of PTHr indicate the compound is probably a full agonist.

Reviewer #2 (Expert in endocrinology; Remarks to the Author):

Tamura and colleagues describe PCO371, a novel orally active compound that acts as an agonist of PTHr1. In a series of *in vitro* and *in vivo* experiments, they demonstrate that oral PCO371 has similar biochemical and renal effects as PTH injections, particularly in hypoparathyroid rats; notably, the effects of PCO371 are longer lasting than those of PTH. This elegant work may well represent the groundwork for an important future treatment option for hypoparathyroid patients. The key results are that oral PCO371 exhibits PTH-like activity with regard to calcium homeostasis in TPTX rats. Although it was less potent than hPTH(1-34) *in vitro*, the *in vivo* effects were more long-lasting than hPTH(1-34) and hPTH(1-84), consistent with greater bioavailability. The report is

highly novel; although other ligands for the PTHR1 receptor have been reported, their mechanism of action is different (such as prolonged attachment to the receptor). The current mechanism, namely a class B GPCR agonist, is original. The data are of high quality and meticulously well-presented. Statistics were used appropriately and the conclusions are overall robust. Appropriate credit is given to previous work and the abstract, introduction and conclusions are lucid and appropriate.

Suggested areas of improvement are:

- 1) There is no comparison between the effects on urinary calcium of PCO371 vs hPTH(1-84). This is an important endpoint because a reduction in hypercalciuria would represent an important therapeutic advance.
- 1) The Discussion should mention that future longer term rat studies are needed to monitor for the development of osteosarcoma.
- 2) Please specify whether hypercalcemia occurred in any of the oral PCO371 regimens in hypoparathyroid rats (Figure 5c).
- 3) It is theoretically possible that activation of PTHR2 has beneficial CNS effects in hypoparathyroid patients treated with PTH. It may be that this salutary effect is lost with PCO371's selective activation of PTHR1.
- 4) It should be stated more explicitly in the Discussion that PCO371 does not seem to be a potential osteoporosis therapy.
- 5) Although the rat data reported here are very promising, it may be premature to have initiated Phase 1 clinical studies at this point. Perhaps a monkey study would be more appropriate as a next step. It is important to emphasize that the rat skeleton is very different from the human one, as the rat undergoes continuous bone remodeling throughout its life.
- 6) In TPTX rats, please indicate if there is a significant difference in the increase in serum calcium between hPTH(1-84) and any of the PCO371 doses (Fig. 5a).
- 7) p.8, line 188-190: thiazides do not result in insufficient serum calcium levels (please clarify sentence).
- 8) p.8, line 194-195: this statement should be modified because hPTH(1-84) can have effects for up to 24 hours in hypoparathyroid patients (Sikjaer J Bone Miner Res 2013; 28(10): 2232-43).
- 9) p.9 line 217-218. It is incorrect to state that bone formation was stimulated more than bone resorption, because PCO371 in fact led to higher s-CTX than hPTH(1-84) (Fig. 4e).

Reviewer #3 (Expert in bone biology; Remarks to the Author):

This interesting paper reports discovery of a low molecular weight compound that activates the PTHR1 and is sufficiently effective with oral delivery to treat the hypoparathyroidism of the TPTX rat.

The paper is clearly written and the methods explained in commendable detail. The in vivo data is very promising for a candidate drug for hypoparathyroidism. The evidence for activation of the PTHR1 is compelling, and the pharmacokinetics, with a prolonged elevated level in the circulation after oral administration, help to explain the efficacy of this compound in correcting the low calcium and high phosphorus of PTH deficiency. The important points are made also that this efficacy is achieved without increasing urinary calcium, and of special interest and relevance to clinical treatment, is that administration of PCO371 together with active vitamin D increases the efficacy of low doses of the latter.

This is certainly the first orally available low molecular weight compound that fully activates the PTHR1, and as far as this reviewer knows, there is no such full agonist available for any of the class B GPCRs.

Specific comments

1. The lead compound after analysis and refinement of the hits following screening was CH5138335. With further chemistry and selection based on solubility, oral deliverability and potency, PCO371 was chosen. How much improvement in specific biological activity was achieved when arriving at the selection of PCP371?
2. The lead compound, CH5138335, was used for the experiment of Fig 3C, showing effect of 1mM compound when the various receptor mutants were tested. It would be useful to know whether the same results are achieved when PCO371 is used with these mutant receptors, since there are structural differences between PCO371 and CH5138335.
3. Page 4 and Suppl Table 1. The major question that I have concerns the selectivity of PCO471. The data showing that the compound acts independently of the extracellular domain of the receptor is convincing, and the evidence that it likely interacts entirely or partially with transmembrane domain 6, with Pro-415 being crucial. Such an action through a membrane domain makes the question of selectivity among the class B GPCRs particularly important. This question is addressed in Suppl Table 1, where the data was obtained by subcontract to Cerep(France). The data is presented in a way that is not easy to understand - with agonist activity defined as <25% of control agonist at 1 and 10 $\mu\text{mol/l}$ of PCO471. What does the 25% refer to? Is it 25% of the EC50 or of the maximum effect for that agonist? I suggest that the explanation of Suppl Table 1 be improved to make it exactly clear how the data was interpreted. Also state clearly that all agonist experiments were done with a cAMP read-out, and whether the antagonist evaluation was done against an EC50 dose of each agonist.

To Reviewer #1

We greatly appreciate your critical readings and various comments.

1. Major comment 1.

Although the new in vivo observations made with PC0371 are important on their own, the major issue of this manuscript is the total absence of a mechanism by which PC0371 acts on the PTHR to stimulate prolonged calcemic and phosphatemic responses. This is particularly critical given that recent studies (not discussed in this manuscript) showed that certain peptidic PTH analogs (known as LA-PTH) trigger similar Ca and P responses, but also display high affinity for PTHR (in its G-protein coupled or uncoupled states) and stimulate sustained cAMP from internalized LAPTH-bound PTHR complexes (See studies from Gardella and Vilardaga's labs in PNAS 2008, PNAS 2013, NIH, and reviewed in Nat Chem Biol 2014 and Pharm Rev 2015). The 10,000-fold reduced affinity of PC0371 for PTHR (Fig. 1) presumably reflects a very rapid dissociation of the ligand and thus little residence time on the receptor and short-lived cAMP signaling. If this prediction is proved as correct, it would be difficult to reconcile binding and signaling properties of LA-PTH with those of PC0371. The authors should address this issue by additional experiments before this manuscript become a good candidate for Nat Comm.

Answer to comment

As you pointed out, PCO371 induced prolonged calcemic and phosphatemic responses in TPTX rats, but we did not directly examine the involvement of cellular mechanisms by which the compound produces the prolonged responses. As you suggested, PTHR1 has the capacity to form two distinct high-affinity conformational states: one is G-protein-uncoupled conformation (R^0), and the other is G-protein coupled conformation (RG). (Molecular Endocrinology, 22, 156-166, 2008). Whereas PTH-related peptide (PTHrP) binds more weakly to R^0 and exhibits only a transient response (Molecular

Endocrinology, 22, 156-166, 2008, Proc Natl Acad Sci, USA., 105, 16525-16530, 2008, J Bone Miner Res, DOI: 10.1002/jbmr.2811, 2016), PTHrP analogs, such as M-PTH(1-28), M-PTH(1-34), and LA-PTH, bind with greater affinity to R⁰ and produce prolonged calcemic responses *in vivo*. The peptides that bind R⁰ with high affinity are able to produce cAMP signaling responses in PTHR1-expressing cells for a certain amount of time after initially binding to the PTHR1. Previous study also suggested that there is a good correlation between R⁰ binding affinity and the duration of the cAMP response induced by a given PTH ligand after initial binding to PTHR1 (Proc Natl Acad Sci, USA., 105, 16525-16530, 2008).

We therefore examined whether PCO371 produces as sustained a cAMP signaling response in UMR-106 cells as LA-PTH in comparison with hPTH(1-34) (cAMP washout assay). The duration of cAMP signaling-response induced by PCO371 was much shorter than that of hPTH(1-34), whereas LA-PTH, a long-acting PTH analog, showed more prolonged cAMP signaling than hPTH(1-34). These results suggest that, unlike LA-PTH, prolonged calcemic and phosphatemic responses of PCO371 *in vivo* are not due to the prolonged signaling at rat PTHR1. Therefore, the most likely explanation for the prolonged responses of PCO371 in TPTX rats derives from its extended pharmacokinetics, because the half-life of PCO371 in the circulation was more prolonged than that of hPTH(1-34). To describe this observation, we added a new figure in the Supplementary Information (Supplementary Fig.2) showing the duration of cAMP-signaling responses and added new sentences about the experiment in page 5, lines 1-14 in the revised manuscript (page 4, line 24 in the original manuscript), and page 9 lines 9-15 (Page 8, lines 17-19 in the original manuscript). We also cited the above-mentioned three references (Refs. 22, 23, and 24 in the revised reference numbers), and added the methods for the cAMP washout assay in Supplementary Methods.

2. **Minor comment 1**

Experiments with UMR106: It is unclear how 1 μ M of PCO371 is unable to produce cAMP (Suppl. Fig. 2C) and IP, but still stimulates significant stimulation of Fos, BGLAP, RANKL transcripts and inhibition of OPG and SOST (Fig. 2).

Answer to comment

As you pointed out, 1 μ mol L⁻¹ of PCO371 was unable to produce cAMP in UMR-106 cells (Supplementary Fig.1c), but exhibited significant effects on PTH-related gene expressions (Fig.2 a-e). One μ mol L⁻¹ of PCO371 also stimulated Ca release in fetal long-bone assays (Fig 2f.). These differences in sensitivity are probably due to the differences in treatment times. In Supplementary Fig. 1c, the cells were treated for 20 minutes for cAMP. However, in Fig.2, cells and long bones were treated for 6 h and 72 h, respectively. Although there is no clear explanation as yet for the observed results, the *in vitro* potency of PCO371 was increased by prolonging the incubation time.

3. **Minor comment 2**

The authors should note that cAMP data done with COS cells (Fig. 1) do not demonstrate that PCO371 is a full agonist due to the unavoidable presence of spare receptors when cells are transiently transfected. However, data with UMR106 cells expressing low levels of PTHR indicate the compound is probably a full agonist.

Answer to comment

As you suggested, the number of PTHR1 in COS-7 cells transiently transfected with PTHR1 is much higher than in a native cell line, UMR-106. In the original manuscript,

we described PCO371 as a full agonist with the results in COS-7 cells (page 3, lines 16-17 in the original manuscript); however, the results should be discussed based on the data from UMR-106 cells. We therefore modified the relevant sentences in page 3, lines 19-20 in the revised manuscript (page 3, line 16-17 in the original manuscript), and added a new sentence in page 4, lines 20-22 in the revised manuscript (page 4, line 16-17 in the original manuscript) to demonstrate the full agonistic activity of PCO371 with the results in UMR-106 cells.

We hope that these revisions will satisfactorily meet the comments and questions you raised.

To Reviewer #2

We greatly appreciate your critical readings and various comments.

1. Suggested area of improvement 1-1

There is no comparison between the effects on urinary calcium of PCO371 vs hPTH(1-84). This is an important endpoint because a reduction in hypercalciuria would represent an important therapeutic advance.

Answer to comment

According to your suggestion, we added a new figure in Supplementary Information (Supplementary Fig. 5e) to show the urinary Ca data after 4-week treatment of hPTH(1-84) in TPTX rats. To explain this result, we also added new sentences in page 7, lines 14-15 in the revised manuscript (page 6, line 26 in the original manuscript).

2. Suggested area of improvement 1-2

The Discussion should mention that future longer term rat studies are needed to monitor for the development of osteosarcoma.

Answer to comment

In accordance with your suggestion, we added new sentences to describe the necessity of future rat safety studies to assess the potential concern of osteosarcoma in page 10, lines 17-25 in the revised manuscript (page 9, line 7 in the original manuscript).

3. Suggested area of improvement 2

Please specify whether hypercalcemia occurred in any of the oral PCO371 regimens in hypoparathyroid rats (Figure 5c).

Answer to comment

In the experiment shown in Fig. 5c, we measured the Ca levels at 0, 6, and 10 h after administrations of PCO371 on Days 1, 7, 14, 22, and 29. At these time points, no hypercalcemia was observed. To describe this observation, we added new sentences in page 7, lines 18-19 in the revised manuscript (page 6, line 26–page 7 line 1) in the original manuscript).

4. Suggested area of improvement 3

It is theoretically possible that activation of PTHR2 has beneficial CNS effects in hypoparathyroid patients treated with PTH. It may be that this salutary effect is lost with PCO371's selective activation of PTHR1.

Answer to comment

As you pointed out, hPTHr2, which is expressed abundantly in CNS, is activated by PTH, but not by PCO371. Thus, if PTH exerts its beneficial effects on CNS in the treatment of hypoparathyroidism via the direct activation of hPTHr2, the lack of activity of PCO371 on hPTHr2 might be a potential drawback of the compound. However, we could not address this issue in preclinical studies so far, for the following two reasons. First, as shown in Supplementary Fig.1b and also reported by Hoare SR et al. (Endocrinology 140, 4419-4425, 1999), not only PCO371 but also PTH is a poor ligand

for rat PTHR2. Therefore, the TPTX rat is not a suitable animal to detect the difference in the PTHR2 responses mediated by PTH and PCO371. Second, we have no other workable animal models for hypoparathyroidism to evaluate this issue. We hope to clarify this issue in future clinical studies; however, it is premature to discuss it further at present without any reliable data. We therefore decided not to include this issue in this manuscript.

5. Suggested area of improvement 4

It should be stated more explicitly in the Discussion that PCO371 does not seem to be a potential osteoporosis therapy.

Answer to comment

As you pointed out, the results obtained in the present study do not support the potential benefits of PCO371 as an anti-osteoporosis drug. However, it is as yet too early to state whether PCO371 can or cannot be used for osteoporosis in clinical use at this stage, because the efficacy of PCO371 has not been fully evaluated at the different doses and intervals. Moreover, because the physiology of the rat skeleton is different from that of the human skeleton, non-rodent animals, such as dog and monkey, should be used to assess the potential of the compound more accurately. We therefore modified the relevant sentences to describe more explicitly that the results reported in the present study do not validate the potential of PCO371 for osteoporosis, and added a few more sentences about the necessity of additional studies to address this issue in page 10, lines 9-16 in the revised manuscript (page 9, lines 1-7 in the original manuscript).

6. Suggested area of improvement 5

Although the rat data reported here are very promising, it may be premature to have initiated Phase 1 clinical studies at this point. Perhaps a monkey study would be more appropriate as a next step. It is important to emphasize that the rat skeleton is very different from the human one, as the rat undergoes continuous bone remodeling throughout its life.

Answer to comment

As you pointed out, the rat skeleton is different from the human skeleton, and more human-like animal models for studying human bone metabolism, such as monkey models, are needed to bridge the gap between animal studies and clinical trials. In the same line of thought, we examined the effects of PCO371 on normal dogs, since dogs have been used to study the human skeleton because of their extensive basic multicellular unit-based remodeling. In a pharmacological study with normal dogs, serum Ca was increased after treatment with oral administrations of PCO371 as potently as with subcutaneous hPTH(1-34). Moreover, in our one-month toxicology studies in normal dogs, we have demonstrated that PCO371 induced PTH-like effects on Ca and phosphate metabolism, but not any serious side effects. To describe the above-mentioned studies, we added new sentences in page 9, line 21–page 10, line 2 in the revised manuscript (page 8, line 23 in the original manuscript).

7. Suggested area of improvement 6

In TPTX rats, please indicate if there is a significant difference in the increase in serum calcium between hPTH(1-84) and any of the PCO371 doses (Fig. 5a).

Answer to comment

As you suggested, there appear to be statistical differences between the

hPTH(1-84)-treated group and some of the PCO371-treated groups in their calcemic actions. However, we did not conduct formal statistical analyses on the data, because the 2 drugs were dissolved in different vehicles: hPTH(1-34) in phosphate-citrate buffer (pH 6.0), and PCO371 in 10% DMSO/10% Kolliphor EL in 10% hydroxypropyl- β -cyclodextrin/0.752% glycine buffer. To avoid misunderstanding, we decided not to include the results of statistical analyses in Fig. 5a.

8. Suggested area of improvement 7

p.8, line 188-190: thiazides do not result in insufficient serum calcium levels (please clarify sentence).

Answer to comment

According to your suggestion, we modified the sentence in page 8, lines 2-4 in the original manuscript, because this may cause confusion. (page 8, lines 19-20 in the revised manuscript).

9. Suggested area of improvement 8

p.8, line 194-195: this statement should be modified because hPTH(1-84) can have effects for up to 24 hours in hypoparathyroid patients (Sikjaer J Bone Miner Res 2013; 28(10): 2232-43).

Answer to comment

As you pointed out, hPTH(1-84) can have effects for up to 24 hours in hypoparathyroidism (J Bone Miner Res 2013; 28(10): 2232-43). Since the sentence in the original manuscript (page 8, lines 5-9) did not reflect our intention precisely, we modified the sentence in page 8, line 22–page 9, line 2 in the revised manuscript. We also deleted three references (Refs 24-26 in the original manuscript), and cited the above-mentioned reference as Ref. 27 in the revised manuscript.

10. Suggested area of improvement 9

p.9 line 217-218. It is incorrect to state that bone formation was stimulated more than bone resorption, because PCO371 in fact led to higher s-CTX than hPTH(1-84) (Fig. 4e).

Answer to comment

As you pointed out, although the increase in bone mass was more pronounced by intravenous treatment than by oral treatment, PCO371 increased bone-turnover not only by oral treatment but also by intravenous treatment. Since bone mass change is produced by a shift in the balance between bone formation and resorption, we speculate that, in comparison with oral PCO371, intravenous PCO371 stimulated bone formation to a greater extent than it stimulated bone resorption. As you suggested, the original sentence did not describe this point clearly and may cause confusion. We therefore modified the sentence in page 9, lines 1-5 in the original manuscript (page 10, lines 9-16 in the revised manuscript).

We hope that these revisions will satisfactorily meet the comments and questions you raised.

To Reviewer #3

We greatly appreciate your critical readings and various comments.

1. Specific comment 1

The lead compound after analysis and refinement of the hits following screening was CH5138335. With further chemistry and selection based on solubility, oral deliverability and potency, PCO371 was chosen. How much improvement in specific biological activity was achieved when arriving at the selection of PCP371?

Answer to comment

The main purpose of this study is to show the potential of PCO371 as a clinical candidate, not to focus on the lead-to-candidate optimization. We therefore deleted all descriptions of CH5138335 in the manuscript.

2. Specific comment 2

The lead compound, CH5138335, was used for the experiment of Fig 3C, showing effect of 1mM compound when the various receptor mutants were tested. It would be useful to know whether the same results are achieved when PCO371 is used with these mutant receptors, since there are structural differences between PCO371 and CH5138335.

Answer to comment

As you suggested, there are structural differences between the lead compound, CH5138335, and PCO371, so it is necessary to demonstrate that the same results are obtained with the two compounds. Thus we examined the effects of PCO371 on the three mutants (C1, C2, and C3) used in Fig.3c, instead of CH5138335. As we expected, PCO371 gave almost the same results as those obtained with CH5138335. We therefore replaced the original Fig.3c (with the data of CH5138335) with a new Fig. 3c (with the data of PCO371).

3. Specific comment 3

Page 4 and Suppl Table 1. The major question that I have concerns the selectivity of PCO471. The data showing that the compound acts independently of the extracellular domain of the receptor is convincing, and the evidence that it likely interacts entirely or partially with transmembrane domain 6, with Pro-415 being crucial. Such an action through a membrane domain makes the question of selectivity among the class B GPCRs particularly important. This question is addressed in Suppl Table 1, where the data was obtained by subcontract to Cerep(France). The data is presented in a way that is not easy to understand - with agonist activity defined as read-out, and whether the antagonist evaluation was done against an EC₅₀ dose of each agonist.

Answer to comment

As you pointed out, the explanations of Supplementary Table 1 and the methods for the "Selectivity assay" are not always clear and may cause confusion. We therefore added the following explanations for the "Selectivity assay" in Methods (page 29, lines 10-11 in the original manuscript) and moved this part to Supplementary Methods: 1) describing the cell lines used for each assay, 2) clarifying that cAMP production is a readout for all the agonist and antagonist experiments, 3) describing how the EC₅₀ or IC₅₀ of control

agonists or antagonists was calculated, 4) clarifying how we calculated the agonistic and antagonistic activity, and 5) giving definitions of percent ranges of agonistic and antagonistic activity.

We hope that these revisions will satisfactorily meet the comments and questions you raised.

REVIEWERS' COMMENTS:

Reviewer #1 (Remarks to the Author):

The authors responded adequately to this reviewer's concerns. The reason why PCO371 is unable to produce cAMP, but still stimulates significant stimulation of transcripts and remains unclear however. This should be clearly pointed out in the text.

Reviewer #2 (Remarks to the Author):

The authors have satisfactorily addressed the concerns that were raised. There is, however, one minor remaining concern. The statement that IV PCO371 as compared to po PCO371 increases bone formation to a greater extent than bone resorption (p. 10 line 249-250) is not clearly supported by the data. Specifically, in Figure 4d, the higher dose IV PCO371 increases osteocalcin and s-CTX levels to the same extent as does po PCO371. Similarly, in Supplementary Table 3, BFR/BS is increased to a similar extent with both po PCO371 and the higher dose IV PCO371; BRs.R is in fact increased more with high dose IV PCO371 than with po PCO371. This statement (p. 10 line 249-250) should thus be revised. In addition, on p. 10 line 249 "oral PTH" should be replaced with "oral PCO371."

Reviewer #3 (Remarks to the Author):

The paper reports interesting and important data and has been improved by the changes made in response to the various matters raised.

The authors have responded satisfactorily to the points that I queried, and I think also to those of the other referees.

Specific matters:

1. The procedures used in the "selectivity assay" have been clarified satisfactorily.

2. The mechanisms by which PCO371 produces the prolonged response.

This was an important point to raise. The authors have added experiments in which they have pre-incubated UMR 106 cells with PCO 371 and with PTH (1 - 34) and the long acting analogue, LA-PTH, washed the cells and looked at intervals thereafter for adenylyl cyclase activation by using a short incubation with phosphodiesterase inhibitor. Unlike LA-PTH, PCO371 shows no evidence of prolonged response in the washout experiment, and they are justified in proposing that the most likely explanation for the prolonged response to PCO371 in vivo is its extended pharmacokinetics. The method and the rationale of the approach they have used have been clearly explained in the revised manuscript.

3. At 1 μ M PCO371 had no effect on cAMP but increased gene expression.

Authors' response was satisfactory, but in fact there are many examples of effects on later cAMP/PKA-related cellular events at hormone or agonist concentrations that are below those causing significantly increased cAMP. For example

(i) Endocrinology 111;178, 1982 - PTH activated PKA at concentrations lower than those producing detectable change in cAMP ;

(ii) PNAS 74: 3419, 1977; JBC 254:3861, 1979; JBC 253; 8994, 1978; JBC 254: 2077, 1979. Steroidogenic hormones hCG & LH, and ACTH in adrenal, show significant activation of PKA at hormone concentrations producing no measurable total cell cAMP change. This and other work led to conclusion that extremely small changes in cAMP serve to transmit signals arising from activation of the receptor-cyclase complex.

4. There are a couple of minor matters that might be attended to in the manuscript:

(i) Line 167: "the effects were more potent and longer lasting than those of hPTH(1-84) (Fig 5a7b) or hPTH(1-34)(Suppl 5a,b)" . This is mis-use of the word, "potent". Obviously, PCO371 is much less potent, but what is important is that it has a greater maximum effect. A more precise wording would be : "...with a greater efficacy and longer lasting effects.."

2. Line 248: "..increased BMD and bone strength as potently as hPTH(1-34)"

Again , potent is not the appropriate word. PCO371 is much less potent than hPTH(1-34) in this in vivo study, but the highest IV dose of PCO371 achieved effects on BMD and bone strength comparable to those with the single dose of hPTH(1-34).

The above two points relate to understanding of English usage.

3. The authors should check the text carefully at several points for the dose of hPTH(1-34) used. Line 596 indicates 0.9 nmol/Kg - but this is an extremely low dose, slightly less than 4 µg/Kg, which would be unlikely to have any effect in the rat. Fig 4 also indicates a dose of 0.9 nmol/Kg, but Fig 5 has it as 9 nmol/Kg. Line 645 also has 9 nmol/Kg.

To Reviewer #1

We greatly appreciate your critical comment.

4. Remark to the author.

The authors responded adequately to this reviewer's concerns. The reason why PC0371 is unable to produce cAMP, but still stimulates significant stimulation of transcripts and remains unclear however. This should be clearly pointed out in the text.

Answer to comment

In accordance with your and Reviewer#3's comments, we added sentences to discuss the possible explanations for this observation in page 5, lines 119-123 in the revised manuscript (page 4, line 105 in the original manuscript). We also cited three references that Reviewer#3 suggested to support the discussion.

We hope that this revision will satisfactorily meet the comment you raised.

To Reviewer #2

We greatly appreciate your critical readings and comments.

11. Remarks to the Author

The authors have satisfactorily addressed the concerns that were raised. There is, however, one minor remaining concern. The statement that IV PCO371 as compared to po PCO371 increases bone formation to a greater extent than bone resorption (p. 10 line 249-250) is not clearly supported by the data. Specifically, in Figure 4d, the higher dose IV PCO371 increases osteocalcin and s-CTX levels to the same extent as does po PCO371. Similarly, in Supplementary Table 3, BFR/BS is increased to a similar extent with both po PCO371 and the higher dose IV PCO371; BRs.R is in fact increased more with high dose IV PCO371 than with po PCO371. This statement (p. 10 line 249-250) should thus be revised. In addition, on p. 10 line 249 "oral PTH" should be replaced with "oral PCO371."

Answer to comment

In accordance with your suggestion, we rewrote the sentence concerning bone turnover and the balance between bone formation and bone resorption after treatments of IV PCO371 and po PCO371 in page 10, lines 249-252 in the original manuscript (page 10, lines 250-254 in the revised manuscript). As you pointed out, we also corrected "oral PTH" to "oral PCO371 in page 10, lines 246-252 in the original manuscript (page 10, line 252 in the revised manuscript) .

We hope that these revisions will satisfactorily meet the comments you raised.

To Reviewer #3

We greatly appreciate your critical readings and various comments.

4. Specific comment 1

At 1 μ M PCO371 had no effect on cAMP but increased gene expression. Authors' response was satisfactory, but in fact there are many examples of effects on later cAMP/PKA-related cellular events at hormone or agonist concentrations that are below those causing significantly increased cAMP. For example (i) Endocrinology 111;178, 1982 - PTH activated PKA at concentrations lower than those producing detectable change in cAMP ; (ii) PNAS 74: 3419, 1977; JBC 254:3861, 1979; JBC 253; 8994, 1978; JBC 254: 2077, 1979. Steroidogenic hormones hCG & LH, and ACTH in adrenal, show significant activation of PKA at hormone concentrations producing no measurable total cell cAMP change. This and other work led to conclusion that extremely small changes in cAMP serve to transmit signals arising from activation of the receptor-cyclase complex.

Answer to comment

In accordance with your and Reviewer#1's comments, we added sentences to discuss the possible explanations for this observation in page 5, lines 119-123 in the revised manuscript (page 4, line 105 in the original manuscript). We also cited three references that you suggested to support the discussion.

12. Minor point 1

(i) Line 167: "the effects were more potent and longer lasting than those of hPTH(1-84) (Fig 5a7b) or hPTH(1-34) (Suppl. 5a,b)" . This is mis-use of the word, "potent". Obviously, PCO371 is much less potent, but what is important is that it has a greater maximum effect. A more precise wording would be : "...with a greater efficacy and longer lasting effects than those.."

Answer to comment

We modified the sentence in page 5, lines 172-174 in the revised manuscript (page 7, lines 166-168 in the original manuscript), as you suggested.

13. Minor point 2

Line 248: "..increased BMD and bone strength as potently as hPTH(1-34)" Again , potent is not the appropriate word. PCO371 is much less potent than hPTH(1-34) in this in vivo study, but the highest IV dose of PCO371 achieved effects on BMD and bone strength comparable to those with the single dose of hPTH(1-34).

Answer to comment

In accordance with your suggestion, we modified the sentences in page 10, lines 250-254 in the revised manuscript (page 10, lines 246-249 in the original manuscript) We also modified the sentence in Introduction in page 3, lines 62-65 in the revised manuscript (page 3, lines 62-64 in the original manuscript)

14. Minor point 3

The authors should check the text carefully at several points for the dose of hPTH(1-34)

used. Line 596 indicates 0.9 nmol/Kg - but this is an extremely low dose, slightly less than 4 µg/Kg, which would be unlikely to have any effect in the rat. Fig 4 also indicates a dose of 0.9 nmol/Kg, but Fig 5 has it as 9 nmol/Kg. Line 645 also has 9 nmol/Kg..

Answer to comment

In our OVX rat study, we used 0.9 nmol/kg (3.6 µg/kg) of hPTH(1-34), because the peak plasma concentrations (C max) after subcutaneous administration of 0.9 nmol/kg of hPTH(1-34) to rats (about 125 pg mL⁻¹) was estimated to be comparable to that after subcutaneous administration of 20 µg of Forteo (hPTH(1-34)) in human ((130 pg mL⁻¹, *Calcif Tissue Int* 2010 87:485). Although 0.9 nmol/kg is considerably lower than that used in other published rat studies, our experience has shown that daily subcutaneous administration of 0.9 nmol/kg hPTH(1-34) is able to induce significant increase in BMD in OVX rats after 12 weeks. The effectiveness of a lower dose of hPTH(1-34) in OVX rats (2.5 µg/kg) has also been reported by another group (*Calcif Tissue Int* 1992 50:214). In our TPTX studies, however, 0.9 nmol/kg of hPTH(1-34) or hPTH(1-84) failed to induce any changes in serum Ca levels, this is probably due to the shorter half-lives of the peptides in rat than those in human (Tmax of hPTH(1-34): 30 min in human versus 5 min in rat (Table 1), Tmax of hPTH(1-84): 15 min at the first peak and 120 min at the second peak in human versus 5-15 min in rat (*Calcif Tissue Int* 2010 87:485, *J Bone Miner Res* 2013 28:2232, *J Pharmacol Exp Ther* 2009 330:169). We therefore used ten times the doses of hPTH(1-34) or hPTH(1-84) (9 mol/kg) as positive controls for our TPTX rat studies.

We hope that these revisions will satisfactorily meet the comments and questions you raised.